# An optimum rate of microtubule flux for error correction in metaphase spindle

Yao Wang[1,3], Jie Wang[1,3] , Yu-Ru Liu[1,3], Peng-Ye Wang[2], Ping Xie[1,3]

**Accurate chromosome segregation requires efficient corrections of erroneous kinetochore–microtubule attachments during metaphase. However, the detailed mechanisms of how the erroneous attachments can be corrected in the metaphase spindle with the presence of microtubule poleward flux are unclear. To explore the mechanisms and understand the roles the flux plays in the error correction, here we study numerically the correction of various erroneous (merotelic, syntelic, and monotelic) attachments in the metaphase spindle exhibiting the flux. We show that with the effect of kinase Aurora B activity, the erroneous attachments can be corrected efficiently. In contrast, without the effect of Aurora B activity the erroneous attachments cannot be corrected efficiently. More interestingly, we find that an optimum rate of the microtubule poleward flux or an optimum amplitude of the kinetochore oscillation is present, which can result in both the efficient error correction and high mitotic fidelity.**

## Introduction

Faithful chromosome segregation during mitosis is essential for maintaining genomic stability and preventing aneuploidy. This process requires precise coordination between the assembly of the bipolar spindle and the establishment of correct kinetochore–microtubule (kinetochore–MT) attachments in metaphase (Zaytsev & Grishchuk, 2015; Lampson & Grishchuk, 2017; Risteski et al, 2021; Sigmund et al, 2024). Each duplicated chromosome carries two sister kinetochores that must attach to kinetochore MTs (kMTs) emanating from opposite spindle poles, forming the amphitelic (bi-oriented) configuration that ensures equal partitioning of genetic material into the daughter cells. Even a single segregation error can give rise to aneuploidy, which underlies a variety of developmental disorders, spontaneous miscarriages, and cancers (Bakhoum and Compton, 2012; Knouse et al, 2014). Despite the inherent stochasticity of spindle assembly and MT dynamics, chromosome segregation in normal somatic cells

occurs with remarkable fidelity, with error rates estimated at only $10^{-4}$–$10^{-3}$ per chromosome per division (Knouse et al, 2014; van den Bos et al, 2016; Lampson & Grishchuk, 2017; Klaasen & Kops, 2022). In contrast, immortalized or cancer-derived cell lines exhibit significantly higher missegregation frequencies, reflecting defects in the mechanisms that safeguard mitotic accuracy.

Such extraordinary fidelity implies the existence of robust error-correction mechanisms that detect and eliminate improper kinetochore–MT attachments such as merotelic, syntelic, and monotelic configurations before anaphase onset. These mechanisms rely on the dynamic turnover of kinetochore–MT interactions, tension-dependent stabilization, checkpoint-mediated surveillance, the phosphorylation by centromeric Aurora A/B kinase, and so on to selectively destabilize erroneous attachments while preserving correct attachments (Salmon & Bloom, 2017; Roscioli et al, 2020; Li et al, 2024; Leça et al, 2025). To explain the error-correction mechanisms, several models have been proposed. One class of models highlights the intrinsic role of kMT dynamics to the efficient correction of attachment errors (Paul et al, 2009; Zaytsev & Grishchuk, 2015; Tanaka & Zhang, 2022). Other models emphasize regulation at the kinetochore–MT interface, particularly through Ipl1/Aurora B kinase–mediated phosphorylation of MT-binding kinetochore proteins such as Ndc80, which modulates the stability of the kinetochore–MT attachments in a phosphorylation-dependent manner (Tubman et al, 2017; Banerjee et al, 2020). An alternative set of models proposed that the mechanical tension generated across properly bi-oriented sister kinetochores serves as a stabilizing cue, whereas the absence of tension promotes detachment of incorrect attachments (Zhang et al, 2013; Mukherjee et al, 2019). In addition, kinetochore geometry has been suggested to limit the probability of the erroneous kMT capture (Magidson et al, 2015; Zaytsev & Grishchuk, 2015; Krivov et al, 2021). A unifying conceptual framework also posits that progressive restriction of attachment, force-dependent attachment lifetime, and destabilization of misaligned attachments constitute the three fundamental principles underlying the error correction (Edelmaier et al, 2020).

In all of abovementioned models for the accuracy of kinetochore–MT attachments, no MT poleward flux was considered,

[1]Laboratory of Soft Matter Physics, Institute of Physics, Chinese Academy of Sciences, Beijing, China    [2]Tsientang Institute for Advanced Study, Hangzhou, China    [3]School of Physical Sciences, University of Chinese Academy of Sciences, Beijing, China

Correspondence: pxie@aphy.iphy.ac.cn

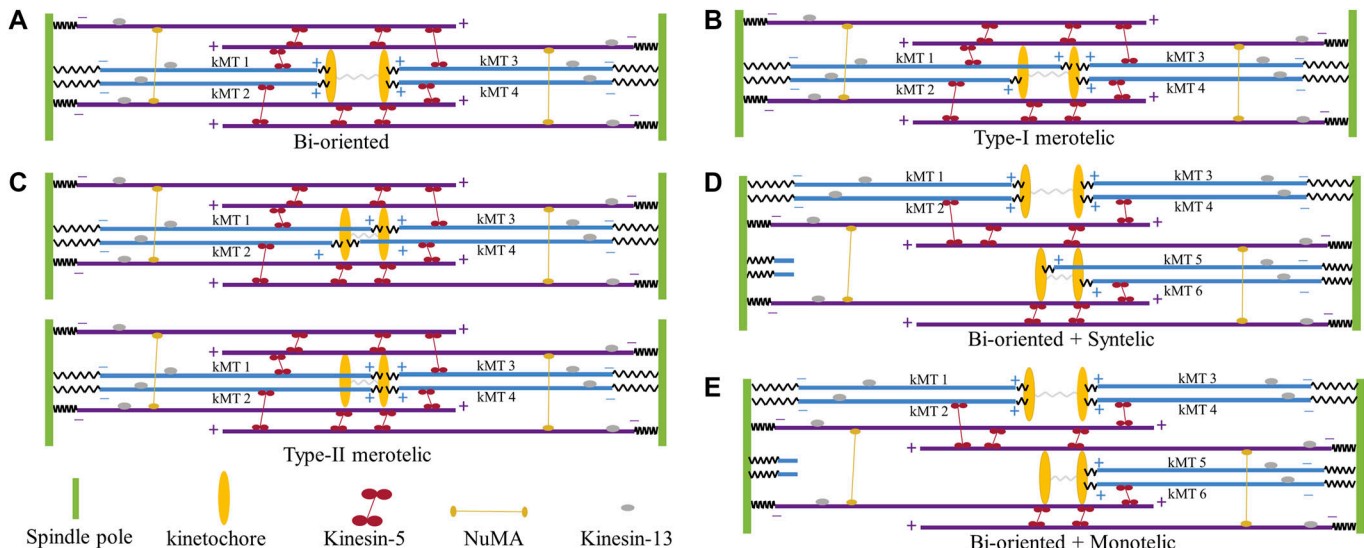

**Figure 1.  Schematic diagrams of the spindle system with correct bi-oriented attachments for *N* = 2 ensembles of MTs and related systems with various erroneous attachments.**
**(A)** Bi-oriented or amphitelic attachments (with no incorrect kinetochore–MT attachment). **(B)** Type-I merotelic attachments (with one incorrect kinetochore–MT attachment). **(C)** Type-II merotelic attachments (with two incorrect kinetochore–MT attachments). **(D)** Syntelic attachments. **(E)** Monotelic attachments. In (D, E), to make the spindle have the stable spindle length in the simulation, besides the syntelic and monotelic attachments, the bi-oriented attachments are also included.

which can be applicable to the spindle of fission yeast *Schizosaccharomyces pombe* (Mallavarapu et al, 1999). However, the MT poleward flux was present in the spindle of higher eukaryotes (Steblyanko et al, 2020; Barisic & Rajendraprasad, 2021; Risteski et al, 2022) and the MT flux could play roles in the correction of erroneous kinetochore–MT attachments. Thus, to study the correction of erroneous attachments in the spindle of higher eukaryotes the MT flux must be considered.

In this work, to study the error correction we modify the previously proposed model of the metaphase spindle with the presence of MT flux (Wang et al, 2025) by considering kinetochore–MT detachments/reattachments. The previous model has successfully explained how the metaphase spindle can automatically maintain its stability in the presence of the MT flux and how the MT-flux rate can regulate the spindle size (Wang et al, 2025). The previous model can also automatically result in the slow and large-amplitude kinetochore oscillations (Wang et al, 2025), the origin of which is due to the stochastic minus-ended MT depolymerization by kinesin-13 KIF2A motors. With the modified model, here we study computationally the correction of erroneous kinetochore–MT attachments. Our numerical results show that Aurora B activity is critical to the efficient error correction. More interestingly, we find that although the low MT-flux rate or the low amplitude of kinetochore oscillation results in a low mitotic fidelity, the high MT-flux rate or the high amplitude of kinetochore oscillation results in the inability of the efficient correction of erroneous merotelic attachments before the anaphase onset. Thus, an optimum MT-flux rate or an optimum amplitude of kinetochore oscillation is present, which can lead to both the high mitotic fidelity and efficient error correction before the anaphase onset.

## Results

### The model

Here, we use the similar model for the dynamics of the metaphase spindle system to that proposed previously (Wang et al, 2025). The difference between the current model and the previous one (Wang et al, 2025) is that in the current model we consider the kMT detachment from and reattachment to the kinetochore, whereas in the previous model, the kMT detachment from the kinetochore was not considered.

For the correct bi-oriented attachments, the spindle system can be shown schematically in Fig 1A. In the model, we do not consider the effect of astral MTs on the dynamics of the spindle, as a lot of experimental data showed (Mitchison et al, 2005; Lacroix et al, 2018). We consider *N* ensembles of MTs in the spindle system, defined as the spindle composed of two kinetochores, two spindle poles, 2*N* kMTs, and *N* pairs of antiparallel interpolar or bridging MTs (bMTs) (namely, 2*N* bMTs) (*N* ≥ 2 being an integer). The two sister kinetochores are connected together by a linker of spring elastic coefficient $\kappa_1$. Each kMT is connected to one kinetochore at the plus end by a linker of spring elastic coefficient $\kappa_2$ and is connected to one spindle pole at the minus end by a linker of spring elastic coefficient $\kappa_3$. Each pair of bMTs can form an antiparallel overlap near their plus ends. For example, Fig 1A shows schematically *N* = 2 ensembles of MTs in the spindle, where there are two sister kinetochores, two spindle poles, 2 × 2 kMTs, and two pairs of antiparallel bMTs (namely, 2 × 2 bMTs). Fig S1A (see the Supporting Information) shows schematically *N* = 3 ensembles of MTs in the spindle, where there are two sister kinetochores, two

spindle poles, 2 × 3 kMTs, and three pairs of antiparallel bMTs (namely, 2 × 3 bMTs).

In the model, three types of proteins are explicitly considered. One type is homotetrameric kinesin-5 motors such as Eg5 that can only bind to MTs in the antiparallel overlap region (van den Wildenberg et al, 2008), with each pair of heads moving toward the plus end to drive the relative sliding of the two antiparallel MTs (Liu et al, 2021). The other type is kinesin-13 motors such as KIF2A that can only bind to the MT regions where the antiparallel overlap is not formed. By making use of the poleward flux of MT and the diffusion of the motor along MT, the kinesin-13 KIF2A motors can arrive at the minus end where they perform the depolymerase activity (McHugh & Welburn, 2023; Xie, 2024). In detail, after reaching the minus end the KIF2A motor together with the pole-localized MT severases uncaps the γ-tubulin ring complex and depolymerizes MT (Henkin et al, 2023). The third type is NuMA proteins that can crosslink two distant parallel bMTs (Harborth et al, 1999), ensuring all bMTs to slide with the nearly same rate and thus ensuring the stability of the spindle system.

The models for the kinesin-5 Eg5, kinesin-13 KIF2A, and NuMA are described before (Wang et al, 2025). For simplicity, the spontaneous polymerization, the polymerization promoted by polymerase enzymes (Reber et al, 2013; Xie, 2023), and the suppression of the MT dynamics by kinesin-8 such as KIF18A motors (Du et al, 2010; Stumpff et al, 2012) are considered to give a mean polymerization rate denoted by $v_{p0}$ at the bMT plus end. This is consistent with the available experimental data showing that in the spindle with the large poleward flux, the MT plus end predominantly grows or pauses (Risteski et al, 2021, 2022, 2024), implying that the depolymerization at the plus end can be neglected. Because the kMT plus end is connected to the kinetochore by linking proteins such as Ndc80 (Wei et al, 2007; Scarborough et al, 2019; Polley et al, 2023), the linking proteins occupy some tubulins near the kMT plus end, and thus, the number of the polymerase enzymes at the kMT plus end is smaller than at the bMT plus end. Moreover, the kinetochore-localized kinesin-13 such as MCAK can contribute to the kMT plus-end depolymerization (Ganem & Compton, 2004; Wordeman et al, 2007; Steblyanko et al, 2020). Thus, when no pulling force is on the kMT plus end, the kMT polymerization rate should be smaller than $v_{p0}$, with the kMT polymerization rate being written as $v_{p0}/B$, where $B > 1$ is a constant. When a pulling force $F$ is on the kMT plus end, which arises from the stretching of the linker connecting the kMT and kinetochore, the kMT polymerization rate can be accelerated (Gudimchuk et al, 2020; Long et al, 2020), which can be simply written as

$$v_{pol}^{(kMT)} = \frac{v_{p0}}{B}\left(1 + \frac{F}{F_{p0}}\right), \tag{1}$$

where $F_{p0}$ is the force–sensitivity parameter. Here, $v_{p0}$ is taken as a preset value, which in the experiments can be varied by the depletion of the plus-end tracking proteins, such as polymerase enzymes, kinesin-8 motors (Stumpff et al, 2008, 2012; Steblyanko et al, 2020; Risteski et al, 2022, 2024). Unless otherwise pointed out, throughout we take $v_{p0}$ = 20 nm/s, resulting in the MT-flux rate to be about 20 nm/s that is consistent with the experimental value in the normal human spindle measured by Risteski et al (2022). After the kMT detachment from the kinetochore, the polymerization rate

of kMTs changes to $v_{p0}$. After the kMT reattachment to the kinetochore, the polymerization rate of kMTs changes back to $v_{pol}^{(kMT)}$, which is calculated by Equation (1). After a kinesin-13 KIF2A motor reaching the minus end of an MT, it can stay there for a time period $\tau_{end}$, during which it can remove the tubulins processively with a rate $k_{dep}$ (Xie, 2024).

Besides the correct bi-oriented attachments (Figs 1A and S1A), we also consider various erroneous attachments, including merotelic (Figs 1B and C and S1B–D), syntelic (Figs 1D and S1E), and monotelic attachments (Figs 1E and S1F).

In the model, the rate of kMT detachment from the kinetochore is considered as follows:

(i) The available experimental results showed that the lifetime of kinetochore attachment in the spindle of the higher eukaryote (Kabeche & Compton, 2013) is similar to that in the yeast spindle (Akiyoshi et al, 2010). Because the force dependence of the kinetochore attachment lifetime in the yeast spindle is available (Akiyoshi et al, 2010), whereas in the spindle of the higher eukaryote, it is unavailable, for the simulation, we consider the force dependence of kinetochore attachment lifetime in the spindle of the higher eukaryote having the same form as that in the yeast spindle. Thus, in the absence of Aurora B, the detachment rate, $r_{detach}$, is equal to the inverse of the kMT attachment time to the kinetochore, $\tau_{attach}$, measured experimentally by Akiyoshi et al (2010). As shown in Fig S2, the dependence of $\tau_{attach}$ on the pulling force $F$ acting on the kMT plus end can be described by the following phenomenological equation:

$$\tau_{attach} = (r_{detach})^{-1} = \left\{k_1\exp\left(\frac{F}{F_2}\right)\left[1-\exp\left(-\frac{F}{F_1}\right)\right] + k_2\exp\left(\frac{F}{F_2}\right)\exp\left(-\frac{F}{F_1}\right)\right\}^{-1}, \tag{2}$$

where $k_1 = 6.6 \times 10^{-4}$ min$^{-1}$, $k_2 = 7.7 \times 10^{-2}$ min$^{-1}$, $F_1 = 1.07$ pN, and $F_2 = 1.86$ pN. From Fig S2, it is seen that the lifetime of the attachment exhibits the catch–slip bonding characteristic, having the maximum value of about 50 min under the pulling force of 5 pN.

(ii) In the presence of Aurora B, to be consistent with the experimental data (DeLuca et al, 2006; Sarangapani et al, 2013; Kalantzaki et al, 2015; Sen et al, 2021; Britigan et al, 2022; Wang et al, 2024), the rate of kMT detachment from the kinetochore, $k'_{detach}$, can be written as

$$k'_{detach} = X_{detach}r_{detach}, \tag{3}$$

$$X_{detach} = 1 + A\exp[-b(x - x_0)], \tag{4}$$

where $k_{detach}$ is the detachment rate in the absence of Aurora B, which is calculated by Equation (2), $X_{detach}$ is the time of the detachment rate enhanced by the effect of Aurora B activity relative to that in the absence of Aurora B, $x$ is the distance between the two sister kinetochores under the force on them, $x_0$ is the equilibrium distance between the two sister kinetochores, namely, the distance between the two sister kinetochores under no force on them (we take $x_0$ = 0.5 μm), 1 + $A$ corresponds to the time of the detachment rate increased by Aurora B activity when the two sister

kinetochores have the equilibrium distance, and $b$ is the factor characterizing the increasing rate of the effect of Aurora B activity on the detachment rate. Here, for simplicity, we consider the effect of Aurora B localizing to the inner centromere (Hindriksen et al, 2017; Yamamoto, 2021) and do not consider the effect of Aurora B localizing to other regions (Broad et al, 2020). To be consistent with the published experimental data showing a decrease in kMT turnover by up to a factor of 65 in the presence of Aurora B inhibitors (Cimini et al, 2006; Edelmaier et al, 2020), we take $1 + A = 65$ or $A = 64$. We take $b$ as a variable parameter.

The rate of kMT attachment to a kinetochore can be approximately calculated as

$$k_{attach} = k_{attach}^{(0)} \exp(-\alpha \kappa_2 |y|), \tag{5}$$

where $y$ is the distance between the plus end of the unattached kMT and a kinetochore, $k_{attach}^{(0)}$ is the attachment rate under $y = 0$, $\kappa_2$ is the spring elastic coefficient of the linker connecting the kMT to the kinetochore (defined above), and $\alpha$ is a factor characterizing the distance or force dependence of the kMT attachment rate. Note that Equation (5) is similar to that used before for the MT detachment rate (Grill et al, 2005). Here, we take $k_{attach}^{(0)}$ and $\alpha$ as variable parameters.

We take the same parameter values for the kinesin-5 Eg5 motor, for the kinesin-13 motor, for the NuMA protein, and for the polymerization activity at the kMT plus end as those used before (Wang et al, 2025) to reproduce quantitatively the experimental data of Risteski et al (2022) (see Tables S1, S2, S3, and S4). We also take the same values of the elastic constant of spring connecting the kinetochore and the plus end of each kMT, $\kappa_2 = 0.1$ pN/nm, and the elastic constant of spring connecting the spindle pole and the minus end of each MT, $\kappa_3 = 0.1$ pN/nm, as those used before (Shimamoto et al, 2011; Wang et al, 2025) (see Table S5). To make the numerical results on the periods of the large-amplitude kinetochore oscillation (see Fig S3A in the Supporting Information) and pulsation (see Fig S3B in the Supporting Information) be similar to the available experimental data (Dumont et al, 2012), we take the elastic constant of spring connecting the two sister kinetochores, $\kappa_1 = N\kappa$, where $\kappa = 0.01$ pN/nm (see Table S5). We use the Monte Carlo method to make the simulation (see the Materials and Methods section).

### Bi-oriented kinetochore–MT attachments have a larger lifetime than erroneous attachments

As it is noted, the correction of erroneous attachments relies on the kMT detachment from the kinetochore. From Equations (2), (3), and (4), it is seen that in the absence of Aurora B the kMT detachment rate depends on the pulling force, $F$, acting on the kMT plus end and the effect of Aurora B activity on the kMT detachment rate depends on the change of the interkinetochore distance, $x–x_0$. Thus, it is interesting to study $F$ and $x–x_0$ in the spindle system with the sister kinetochores having correct and various erroneous attachments, where the kMTs cannot detach from the kinetochore.

In Fig 2A, we show the simulated results of $F$ and $x–x_0$ for the types of attachments shown in Fig 1A–C, where $F$ and $x–x_0$ represent

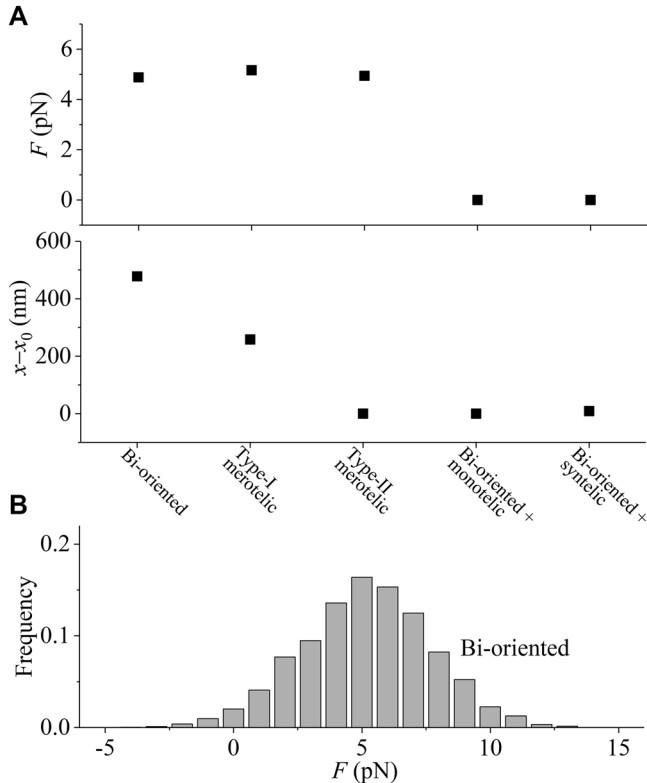

**Figure 2. Results for pulling force on the plus end of kMT and interkinetochore distance in the spindle with kMT attached fixedly to the kinetochore.**
**(A)** Time-averaged values of the pulling force, $F$, acting on each kMT plus end (upper panel) and the change of the interkinetochore distance, $x–x_0$ (lower panel), in the spindle systems with the sister kinetochores having correct and various erroneous attachments shown in Fig 1A–E, where the kMTs cannot detach from the kinetochore. Each data point is calculated from 20,000 recorded values, sampled at 1-s intervals. **(B)** Distribution of the force exerted on the plus end of each kMT for the bi-oriented attachments, which is analyzed using 20,000 recorded force values, sampled at 1-s intervals. The pulling forces are defined as positive, and the pushing forces from the MT plus end toward the minus end are defined as negative. Note that this numerical distribution resembles well the published experimental one (Mukherjee et al, 2019), with both having similar half-width and similar force where the maximum frequency occurs.

the time-averaged values. It is evident that for the syntelic and monotelic attachments shown in Fig 1D and E, we have $F = 0$ and $x–x_0 = 0$, which are also shown in Fig 2A. From Fig 2A, it is seen that for the bi-oriented and merotelic attachments, where kMTs emanating from one spindle pole have the same number as those from another spindle pole, the pulling force $F$ on each kMT plus end has the mean value of about 5 pN (see also Fig 2B), under which the kMT has the maximum attachment time to the kinetochore (see Fig S2). In contrast, for the syntelic and monotelic attachments, where all kMTs connected to the two kinetochores emanate from the same spindle pole, the pulling force $F$ on each kMT plus end has the mean value of zero, under which the kMT has the shorter attachment time. From Fig 2A, it is more interesting to see that for any type of the erroneous attachments the change of the interkinetochore distance, $x–x_0$, has a smaller value than for the correct bi-oriented attachments. Because the kMT detachment rate

increases exponentially with the decrease of $x–x_0$, as seen from Equations (3) and (4), the above results thus imply that Aurora B activity has a large effect on the enhancement of the kMT detachment from the kinetochores having the erroneous attachments than that from the kinetochores having the correct attachments.

The results shown in Fig 2A are for the spindle systems shown in Fig 1A–E, with the corresponding system that has the equal number of MTs emanating from the two opposite spindle poles consisting of $N = 2$ ensembles of MTs. For the spindle systems shown in Fig S1A–F, with the corresponding system that has the equal number of MTs emanating from the two opposite spindle poles consisting of $N = 3$ ensembles of MTs, we have the similar results (see Fig S4 in the Supporting Information). From Fig S4, we also see that for the bi-oriented and merotelic attachments the pulling force $F$ on each kMT plus end has the mean value of about 5 pN, whereas for the syntelic and monotelic attachments, the pulling force $F$ on each kMT plus end has the mean value of zero. From Fig S4, it is also interesting to see that for any type of erroneous attachments the change of the interkinetochore distance, $x–x_0$, has a smaller value than for the correct bi-oriented attachments.

Taken together, we show that with the catch–slip bonding interaction of the kMT with the kinetochore (Fig S2) and the effect of Aurora B activity, only the attachment in the spindle having the correct bi-oriented attachments has the maximum lifetime, whereas the attachment in the spindle having erroneous attachments has the shorter lifetime (see, e.g., Fig S5 in the Supporting Information). This implies that in the spindle having the erroneous attachments the kMT can detach easier from the kinetochore and the spindle having the correct bi-oriented attachments can maintain stably for a long time. This thus ensures the highly efficient correction of the erroneous attachments, with the spindle maintaining the high fidelity. The fidelity is defined here as the probability of the spindle being in the state with only the correct bi-oriented attachments among the states with the correct bi-oriented attachments and the states with erroneous attachments.

## Correction of merotelic attachments

In this section, we study the correction of merotelic attachments (Fig 1B and C), where kMTs emanating from one spindle pole have the same number as those from another spindle pole.

First, we consider the spindle system composed of $N = 2$ ensembles of MTs (Fig 1A–C). We begin our simulation with the spindle system in the type-II merotelic state (Fig 1C). During the simulation, any kMT (either correct or incorrect attachment) can detach from and reattach to the kinetochore, with the rates being calculated by Equations (2), (3), (4), and (5). It is evident that the three states, including bi-oriented state (Fig 1A), type-I merotelic state (Fig 1B), and type-II merotelic state (Fig 1C), can occur during the simulation. In Fig 3A, we show one example for the temporal evolution of the occurrences of the three states, which is simulated by taking $b = 50 \ \mu m^{-1}$, $k_{attach}^{(0)} = 100 \ s^{-1}$, and $\alpha = 0.05 \ pN^{-1}$. In Fig S6 (see the Supporting Information), we show another example. It is seen that in about 5.1 min (306 s) in Fig 3A and about 0.9 min (54 s) in Fig S6, the erroneous attachments transition to the correct bi-oriented attachments for the first times. The difference of the transition time for the first times in Figs 3 and S6 is due to the stochasticity of

the system or the simulation. As expected, before the first transitioning to the bi-oriented state, type-I merotelic attachments can also occur (Figs 3 and S6). After the first transitioning to the bi-oriented state, the spindle system maintains almost at the bi-oriented state, although the system can occasionally transition back to type-I merotelic state and the kMT detachment can occur frequently (Figs 3 and S6). If the type-I merotelic state or kMT detachment occurs, the system can return rapidly to the bi-oriented state.

After the first transitioning to the bi-oriented state, the probability of the spindle system maintaining the bi-oriented state (denoted by $P_{bi}$) is about 0.996 (Fig 3B), which is calculated with a long simulation time of 333 min (20,000 s). Here, $P_{bi}$ is defined as the time period of the bi-oriented attachments divided by the total time period of the bi-oriented and all erroneous attachments, namely, defined as the time period of the bi-oriented attachments divided by the total time period without inclusion of the time period of the temporary kMT detachment, because the detached kMT can reattach rapidly (with a mean time of about 0.044 s; see Fig 3). This value of $P_{bi}$ indicates that the probability of the spindle in the bi-oriented state at the moment when the transition from the metaphase to anaphase occurs is about 0.996. This implies a high mitotic fidelity of 0.996 for a pair of chromosomes per cell division, with the erroneous-attachment probability of only 0.4%. For a human spindle with 46 chromosomes, a simple estimate gives $46 \times 0.4/2 \approx 9.2$ times of missegregate chromosomes per 100 cell divisions. Note here that we have not considered the geometric constraints in our one-dimensional model, the inclusion of which can further reduce the erroneous attachments. Considering that the inclusion of the geometric constraints can further reduce the erroneous attachments by several times (e.g., four times) (Edelmaier et al, 2020), the above estimate gives about 2.3 times of missegregate chromosomes per 100 cell divisions, which is consistent with ~0.5–5 times of missegregate chromosomes per 100 cell divisions in normal cells (Klaasen & Kops, 2022). Therefore, our results indicate that the erroneous merotelic attachments (Fig 1B and C) can be efficiently corrected.

The mechanism of efficient error correction can be explained as follows. In the spindle with the erroneous attachments, because of the low attachment time of a kMT to the kinetochore, a kMT can detach from the kinetochore rapidly. In the type-II merotelic state (e.g., the upper panel of Fig 1C), if one correctly attached kMT (kMT 2 or kMT 3) detaches from the kinetochore, because the detached kMT is closer to the kinetochore from which it has just detached than to the other kinetochore, the detached kMT prefers to reattach rapidly to the former kinetochore. If one incorrectly attached kMT (e.g., the left kMT 1) detaches from the right kinetochore, the pulling force on the left attached kMT 2 would increase, whereas the pulling forces on the two right attached kMT 3 and kMT 4 would decrease. Thus, from Equation (1) it is noted that the polymerization rate of the left kMT 2 would increase, whereas the polymerization rates of the right kMT 3 and kMT 4 would decrease, resulting in the two kinetochores to move rightward. This thus increases the probability of the detached kMT 1 to attach to the left kinetochore, whereas this decreases the probability of the detached kMT 1 to reattach to the right kinetochore, because the detached kMT 1 has the nearly same length as that before its

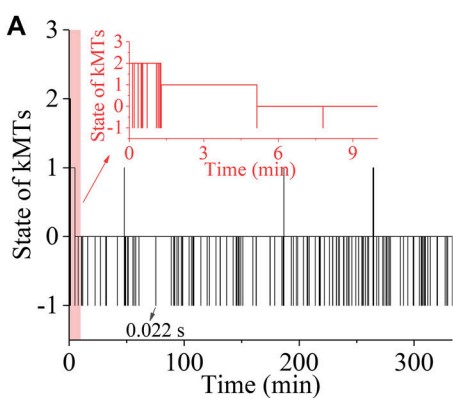 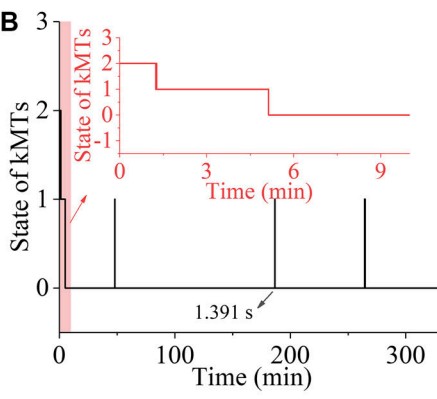

**Figure 3. Temporal evolution of the occurrences of the type-I merotelic state, type-II merotelic state, and bi-oriented state in the spindle system composed of $N = 2$ ensembles of MTs.**
The simulations begin with the spindle system in the type-II merotelic state (Fig 1C). **(A)** Results including the occurrences of the type-I merotelic state, type-II merotelic state, bi-oriented state, and kMT detachment (with the inset showing the results in the region between 0 and 10 min). When the state with no incorrect attachment (bi-oriented attachments) occurs, it is set as the value equal to "0"; when the state with one incorrect attachment (type-I merotelic attachments) occurs, it is set as the value equal to "1"; when the state with two incorrect attachments (type-II merotelic attachments) occurs, it is set as the value equal to "2"; and the state with the occurrence of kMT detachment is set as the value equal to "−1." During the period after the spindle reaching the bi-oriented state for the first times, that is, after about 5.1 min or 306 s, if the kMT detachment from the kinetochore occurs, the average time for the spindle to return to the bi-oriented state is about 0.044 s. One example for the time (0.022 s) required for the state with the occurrence of kMT detachment to return to the bi-oriented state is indicated by the black arrow. **(B)** Same results as in (A) but with exclusion of kMT detachment (with the inset showing the results in the region between 0 and 10 min).

detachment. Hence, the detached kMT 1 prefers to attach to the correct left kinetochore. Similarly, in Fig 1B after the left incorrectly attached kMT 1 detaches from the right kinetochore, the probability of the detached kMT 1 to attach to the left kinetochore increases, whereas the probability of the detached kMT 1 to reattach to the right kinetochore decreases, facilitating transition to the correct bi-oriented attachments. Moreover, the lifetime of the correct bi-oriented attachments in Fig 1A that is longer than those of the erroneous attachments in Fig 1B and C is another factor to ensure the high mitotic fidelity.

Now, we study how the three parameters $b$, $k_{attach}^{(0)}$, and $\alpha$ affect the correction of the merotelic attachments. We still begin our simulations with type-II merotelic attachments (Fig 1C). Our simulations show that for a given $b$, provided that $k_{attach}^{(0)}$ and $\alpha$ are in the range above the lines shown in Fig 4A, the spindle system can maintain stably, with the probability of the correct bi-oriented state, $P_{bi}$, having the mean value of about 0.992. In addition, it is noted that as $k_{attach}^{(0)}$ increases, the mean first time for type-II merotelic state to transition to the bi-oriented state, which is denoted by $\tau$, is increased (Fig 4B). Thus, to have the fast and effective correction of erroneous attachments, $k_{attach}^{(0)}$ should have the value in the range above the lines shown in Fig 4A, but $k_{attach}^{(0)}$ should not be very large. In other words, $k_{attach}^{(0)}$ should have the value that is above but not far from the lines shown in Fig 4A. From Fig 4, we note that the values of $b = 50~\mu m^{-1}$, $k_{attach}^{(0)} = 100~s^{-1}$, and $\alpha = 0.05~pN^{-1}$, as used in Fig 3, are appropriate ones that can give the fast and effective correction of erroneous attachments. Thus, unless otherwise stated, throughout this work we fix these values of $b$, $k_{attach}^{(0)}$, and $\alpha$.

Then, we consider the spindle system composed of $N = 3$ ensembles of MTs (Fig S1A–D). We begin our simulation with the spindle system in the type-III merotelic state (Fig S1D). It is evident that the four states, including bi-oriented state (Fig S1A), type-I merotelic state (Fig S1B), type-II merotelic state (Fig S1C), and type-III merotelic state (Fig S1D), can occur during the simulation. In Fig 5, we show one example for the temporal evolution of the occurrences of the four states, which is simulated using the same

values of $b = 50~\mu m^{-1}$, $k_{attach}^{(0)} = 100~s^{-1}$, and $\alpha = 0.05~pN^{-1}$ as used in Fig 3. It is seen that in about 2.6 min (156 s), the erroneous attachments transition to the correct bi-oriented attachments for the first times. Then, the spindle system maintains almost at the bi-oriented state, albeit occasionally transitioning back to the type-I merotelic state and then returning rapidly to the bi-oriented state. After transitioning to the bi-oriented state for the first times, the time-averaged probability of the system maintaining the bi-oriented state, $P_{bi}$, is 0.996, which is calculated with a long simulation time of 333 min (20,000 s), implying that the system is almost in the correct bi-oriented state. In other words, the erroneous merotelic attachments can be corrected efficiently. These results for $N = 3$ ensembles of MTs (Fig 5) are similar to those for $N = 2$ ensembles of MTs (Figs 3 and S6), indicating that our conclusion is independent of $N$. Therefore, in the following studies, to save the simulation time we focus on $N = 2$ ensembles of MTs.

Taken together, we show that the merotelic attachments can be corrected efficiently, with the finally stable kinetochore–kMT attachments having the mean fidelity of about 0.992, indicating that the probability of the spindle in the bi-oriented state at the moment when the transition from the metaphase to anaphase occurs is about 0.8% for a pair of chromosomes per cell division. This is equivalent to about 46 × 0.8/2 = 18.4 times of missegregate chromosomes per 100 cell divisions in a human spindle with 46 chromosomes. Because the inclusion of the geometric constraints, which have not been considered in our simulation, can further reduce the erroneous attachments by several times (e.g., four times) (Edelmaier et al, 2020), the above estimate gives about 4.6 times of missegregate chromosomes per 100 cell divisions, which is consistent with the experimental value of 0.5–5 times in normal cells (Klaasen & Kops, 2022).

## Correction of monotelic and syntelic attachments

In this section, we focus on the syntelic and monotelic attachments (Fig 1D and E), where all kMTs connected to the pair of kinetochores emanate from the same spindle pole (e.g., the right pole). We

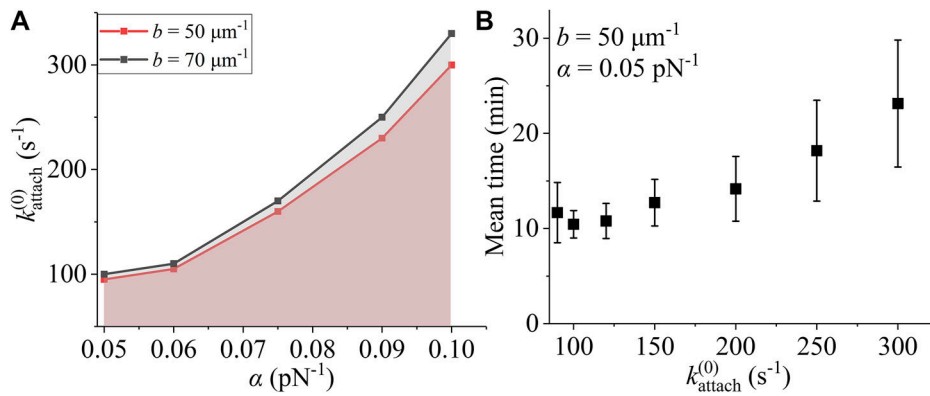

**Figure 4. Effects of parameters $b$, $k_{attach}^{(0)}$, and $\alpha$ on the correction of merotelic attachments in the spindle system composed of $N$ = 2 ensembles of MTs.**
The simulations begin with the spindle system in the type-II merotelic state (Fig 1C). **(A)** Lines are the relationship between $k_{attach}^{(0)}$ and $\alpha$ for $b$ = 50 and 70 $\mu m^{-1}$, above which the spindle system can maintain stably with the probability of the correct bi-oriented state, $P_{bi}$, having the mean value of about 0.992 for a simulation time of at least 333 min (20,000 s). **(B)** Mean first time (mean ± SEM, with 20 records) for the merotelic state to transition to the bi-oriented state, $\tau$, versus $k_{attach}^{(0)}$. Note that $\tau$ is insensitive to the variation of $b$ in the range between 50 and 70 $\mu m^{-1}$ and the variation of $\alpha$ in the range between 0.05 and 0.1 $pN^{-1}$.

consider two cases. One case (case I) is for the treated cell, where for the pair of kinetochores with the syntelic and monotelic attachments, no corresponding kMTs can grow from the opposite (left) pole. The other case (case II) is for the normally untreated cell, where for the pair of kinetochores with the syntelic and monotelic attachments, the corresponding kMTs can grow from the opposite (left) pole. To ensure the spindle having the stable spindle length in the simulation, besides the pair of kinetochores with the syntelic and monotelic attachments, we also include the pair of kinetochores with the bi-oriented attachments in the spindle shown in Fig 1D and E.

First, consider case I. We begin our simulation with the spindle in the monotelic attachments, where the pair of kinetochores is initially positioned at the middle region of the spindle (Fig 1E). During the simulation, any kMT can detach from and reattach to the kinetochores, with the rates being calculated by Equations (2), (3), (4), and (5). We use the same values of $b$ = 50 $\mu m^{-1}$, $k_{attach}^{(0)}$ = 100 $s^{-1}$, and $\alpha$ = 0.05 $pN^{-1}$ as used in Fig 3. Because for both monotelic and syntelic attachments the change of the interkinetochore distance, $x$–$x_0$, has a mean value of zero (Fig 2A), the detachment of a kMT from and reattachment of the detached kMT to a kinetochore and thus the transitions between the syntelic and monotelic states can occur frequently, as seen from Fig 6A (left panel), where we show an example of our simulation results for the temporal evolution of the occurrences of the syntelic and monotelic states. Correspondingly, the temporal evolutions of the two kMT lengths are also shown in Fig 6A (right panel). From Fig 6A (right panel), it is seen that as time evolves, the kMT lengths decrease until reaching the short equilibrium length, under which the mean depolymerization rate at the minus ends becomes equal to the polymerization rate, $v_{pol}^{(kMT)} = v_{p0}/B$, at the plus ends. Then, the kMT lengths fluctuate around the short equilibrium length, which is consistent with the available experimental evidence that monotelic attachments resulted in the positioning of the chromosome close to the spindle pole to which it is attached (Cimini, 2008). Note that during all of the simulations, the bi-oriented kMT attachments in the system maintain stable.

Second, consider case II. The simulation procedure for case II is the same as that for case I. Initially, the length of the two kMTs from the left pole is set as zero. The two kMTs from the left pole can be

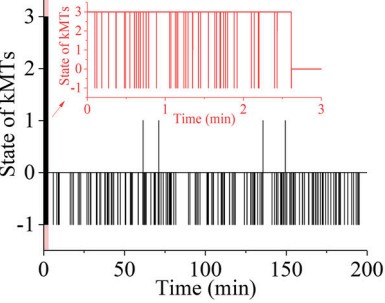

**Figure 5. Temporal evolution of the occurrences of the type-III merotelic state, type-II merotelic state, type-I merotelic state, and bi-oriented state in the spindle system composed of $N$ = 3 ensembles of MTs.**
The inset shows the results in the region between 0 min and 3 min. The simulations begin with the spindle system in the type-III merotelic state (Fig S1D). When the state with no incorrect attachment (bi-oriented attachments) occurs, it is set as "0"; when the state with one incorrect attachment (type-I merotelic attachments) occurs, it is set as "1"; when the state with two incorrect attachment (type-II merotelic attachments) occurs, it is set as "2"; when the state with three incorrect attachment (type-III merotelic attachments) occurs, it is set as "3"; and the state with the occurrence of kMT detachment is set as the value equal to "−1."

polymerized at the plus end with velocity $v_{p0}$ and can be depolymerized at the minus end if a kinesin-13 motor is present there. When the two kMTs grow to the length, which can form the antiparallel overlaps with other MTs, the kinesin-5 motors can bind in the region of the newly formed antiparallel overlaps. As stated in the model (see section entitled "The model"), the kinesin-13 motors can only bind to the MT regions where the antiparallel overlap is not formed. Our simulations show that as the two right kMT lengths decrease, the two left kMT lengths increase, as shown in Fig 6B (left panel). After the two right kMT lengths decreases to the minimum value, where the pair of kinetochores is positioned close to the right pole, the two left kMT lengths can still increase (and sometimes decrease) until one of the left kMTs attaches to the left kinetochore, followed rapidly by the attachment of the other left kMT to the left kinetochore. Then, because of the pulling force $F$ on the kMT plus ends and the polymerization rate at the plus ends of the left kMTs decreasing from $v_{p0}$ to $v_{pol}^{(kMT)}$ described by Equation (1), the left two kMT lengths become decreasing, whereas the right

▶ Life Science Alliance

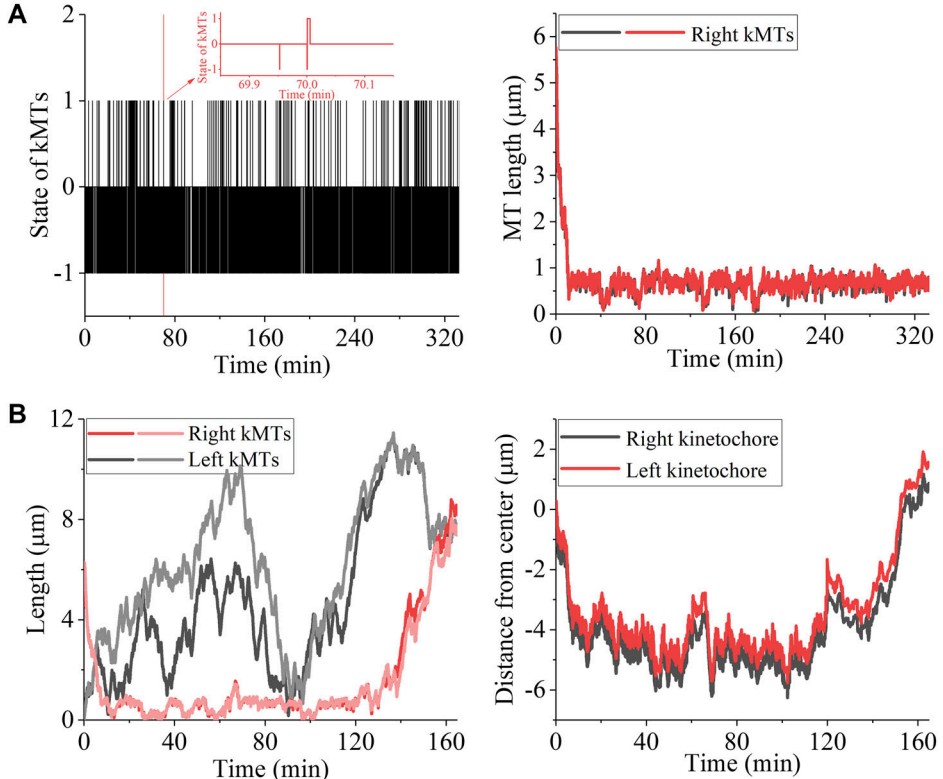

**Figure 6. Correction of syntelic and monotelic attachments.**
The simulations begin with the spindle system in the monotelic state with the pair of kinetochores being initially positioned at the middle region of the spindle (Fig 1E). **(A)** Simulation results for case I, where no corresponding kMTs can grow from the opposite (left) pole. The left panel shows the temporal evolution of the occurrences of syntelic and monotelic states (with the inset showing the results in the region between 69.85 and 70.15 min). When the state with monotelic attachments occurs, it is set as the value equal to "0"; when the state with syntelic attachments occurs, it is set as the value equal to "1"; and the state with the occurrence of kMT detachment is set as the value equal to "–1." The right panel shows temporal evolutions of the two right kMT lengths. Both the left and right panels in (A) correspond to the same simulation run. **(B)** Simulation results for case II, where the corresponding kMTs can grow from the opposite (left) pole. The left panel shows the temporal evolution of the four kMT lengths. The right panel shows temporal evolutions of the positions of the two kinetochores. Both the left and right panels in (B) correspond to the same simulation run.

kMT lengths become increasing. At last, the four kMTs become having the similar lengths, with the two kinetochores fluctuating near the middle region of the spindle and the four kMTs being in the bi-oriented attachments to the two kinetochores (Fig 6B, right panel). As it is noted, before transitioning finally to the bi-oriented attachments the monotelic and syntelic attachments can transition to the merotelic attachments. The mean time for the monotelic or syntelic attachments to transition finally to the bi-oriented attachments is calculated to be about 73.7 ± 24.9 min (with 10 simulations). Because the syntelic and monotelic attachments can be detected by the spindle assembly checkpoint and the time before the anaphase onset can be elongated (Cimini, 2008), our results imply that in the finally stable state, all of the erroneous monotelic and syntelic attachments can be corrected.

Taken together, we show that in the treated cell, where for the pair of kinetochores with the syntelic and monotelic attachments no corresponding kMTs can grow from the opposite pole, as time evolves, the kMT lengths decrease until reaching the short equilibrium length, around which the kMT lengths fluctuate. This explains the available experimental results showing that the monotelic attachments resulted in the positioning of the chromosome close to the spindle pole to which it is attached (Cimini, 2008). In the normally untreated cell, where for the pair of kinetochores with the syntelic and monotelic attachments the corresponding kMTs can grow from the opposite pole, the syntelic and monotelic attachments can be corrected finally to the bi-oriented attachments.

## Aurora B activity is essential for error correction

In the above studies, we considered the effect of Aurora B activity. We see that with Aurora B activity, the merotelic attachments can be corrected efficiently and the syntelic/monotelic attachments can be corrected. In this section, we consider the case without Aurora B activity.

We begin our simulation with the spindle system in the type-II merotelic state (Fig 1C), where the rates of kMT detaching from and reattaching to the kinetochore are still calculated by Equations (2), (3), (4), and (5) but with Equation (4) being replaced with $X_{detach}$ = 1. We use the same values of $k_{attach}^{(0)}$ = 100 s$^{-1}$ and $\alpha$ = 0.05 pN$^{-1}$ as used in Fig 3. In Fig 7A, we show one example for the temporal evolution of the occurrences of the type-I merotelic state, type-II merotelic state, and bi-oriented state. It is seen that it takes a long time (about 209 min) for the merotelic attachments to transition to the correct bi-oriented attachments for the first times. This implies that during the metaphase with the period being shorter than 209 min, namely, before the anaphase onset, there is only a small probability of the merotelic attachments that can transition to the correct bi-oriented attachments. From 10 simulation trajectories, we obtain statistically the probability of the merotelic attachments that can transition to the correct bi-oriented attachments versus the simulation time, as shown in Fig 7B. It is seen that for the long simulation time of 100 min, there is only about 57% probability of the merotelic attachments that can transition to the correct bi-oriented attachments. Even for the very long time of 200 min, there is only about 69% probability of the merotelic attachments that

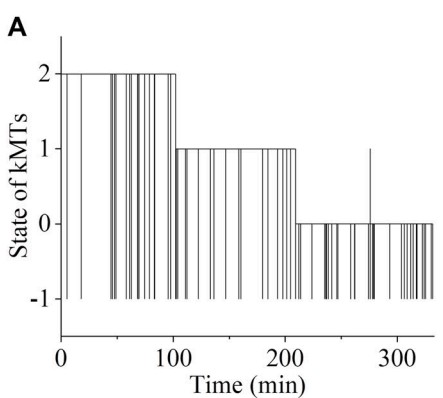

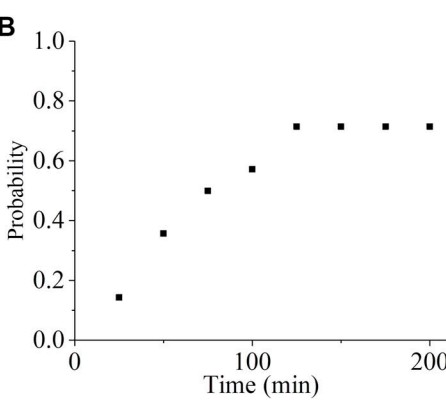

**Figure 7. Effect of Aurora B activity on error correction.**
**(A)** Trajectory for the temporal evolution of the occurrences of the type-I merotelic state, type-II merotelic state, and bi-oriented state in the spindle composed of $N$ = 2 ensembles of MTs without the effect of Aurora B activity. The simulations begin with the spindle system in the type-II merotelic state (Fig 1C). When the state with no incorrect attachment (bi-oriented attachments) occurs, it is set as "0"; when the state with one incorrect attachment (type-I merotelic attachments) occurs, it is set as "1"; when the state with two incorrect attachments (type-II merotelic attachments) occurs, it is set as "2"; and the state with the occurrence of kMT detachment is set as the value equal to "–1." **(B)** Probability of the merotelic attachments that can transition to the correct bi-oriented attachments for the first times versus the simulation time.

can transition to the correct bi-oriented attachments. Because the merotelic attachments cannot be detected by the spindle assembly checkpoint (Cimini, 2008), the results of Fig 7 indicate that without the effect of Aurora B activity, the erroneous merotelic attachments cannot be corrected efficiently before the anaphase onset. The low probability of the merotelic attachments that can transition to the correct bi-oriented attachments within a given long time is due to the same low detachment rate of the incorrectly and correctly attached MTs, because without the effect of Aurora B activity, the attachment lifetime is independent of the inter-kinetochore distance, and thus, the attachments in Fig 1B and C have the same long lifetime as that in Fig 1A.

Taken together, we show that without the effect of Aurora B activity, the erroneous attachments cannot be corrected efficiently before the anaphase onset. In other words, Aurora B activity is indispensable to the efficient correction of the erroneous attachments.

### MT poleward flux of an optimum rate is critical to error correction

Up to now, we have fixed the kMT-flux rate at about 20 nm/s that is consistent with the experimental value in the normal human spindle (Risteski et al, 2022). In this section, we study the effect of the kMT-flux rate on the error correction with the consideration of Aurora B activity. In our simulation, the variation of the kMT-flux rate is achieved by varying $v_{p0}$, as done before (Wang et al, 2025).

We begin our simulation with the spindle system in the type-II merotelic state (Fig 1C). The rates of kMT detaching from and reattaching to the kinetochore are still calculated by Equations (2), (3), (4), and (5). We use the same values of $b$ = 50 $\mu m^{-1}$, $k_{attach}^{(0)}$ = 100 $s^{-1}$, and $\alpha$ = 0.05 $pN^{-1}$ as used in Fig 3. In Fig 8A, we show one example for the temporal evolution of the occurrences of the type-I merotelic state, type-II merotelic state, and bi-oriented state for a low kMT-flux rate $k_{flux}$ = 6 nm/s. In Fig 8B, we show one example for the temporal evolution of the occurrences of the type-I merotelic state, type-II merotelic state, and bi-oriented state for a high kMT-flux rate $k_{flux}$ = 36 nm/s.

From Fig 8A, it is seen that for the low kMT-flux rate $k_{flux}$ = 6 nm/s, the system can transition frequently between the bi-oriented and merotelic states, giving a low probability $P_{bi}$. Because the kMT-flux

rate is proportional to the amplitude of the kinetochore oscillation (Fig 8E), which is consistent with the available experimental results (Stumpff et al, 2008, 2012), the results of Fig 8A also indicate that the kinetochore oscillation of the small amplitude can result in a low mitotic fidelity. This is consistent with the experimental results observed by Iemura et al, showing that the reduction of chromosome oscillation enhanced the number of erroneous kinetochore–kMT attachments and chromosome missegregation (Iemura et al, 2021). From Fig 8B, it is seen that for the high kMT-flux rate $k_{flux}$ = 36 nm/s, the type-II merotelic attachments cannot transition to the bi-oriented attachments within the long simulation time of about 217 min. Because the merotelic attachments cannot be detected by the spindle assembly checkpoint (Cimini, 2008), the results of Fig 8B indicate that for the high kMT-flux rate, the initially erroneous merotelic attachments cannot be corrected before the anaphase onset. This is consistent with the previous experimental studies, showing that the depletion of kinesin-8 motor, giving an increase of $v_{p0}$ and thus an increase of the kMT-flux rate $k_{flux}$, led to chromosome misalignment (Risteski et al, 2024). In Fig 8C, we show the results of probability $P_{bi}$ versus kMT-flux rate $k_{flux}$ after the erroneous attachments transitioning to the correct bi-oriented attachments for the first times, and in Fig 8D, we show the results of the mean first time ($\tau$) for the erroneous attachments to transition to the correct bi-oriented attachments versus kMT-flux rate $k_{flux}$. From Fig 8C, it is seen that $P_{bi}$ increases with $k_{flux}$ and becomes leveled off at high $k_{flux}$. From Fig 8D, it is seen that $\tau$ has a small value when $k_{flux}$ is smaller than about 22 nm/s and increases rapidly with the increase of $k_{flux}$ when $k_{flux}$ becomes larger than about 22 nm/s. These results indicate that only at the optimum $k_{flux}$ that is around 20 nm/s, as measured experimentally in the normal human spindle (Risteski et al, 2022, 2024), can the merotelic attachments be corrected efficiently before the anaphase onset.

The mechanisms for the above results of $P_{bi}$ and $\tau$ versus $k_{flux}$ or for the presence of an optimum $k_{flux}$ can be explained as follows. First, consider the low $k_{flux}$. In the bi-oriented state (Fig 1A), upon a kinesin-13 motor reaching the minus end of one kMT (e.g., kMT 2) connecting to the left kinetochore, the depolymerase activity of this kinesin-13 motor would result in a large difference in lengths between the two kMTs (kMT 2 and kMT 1) emanated from the same

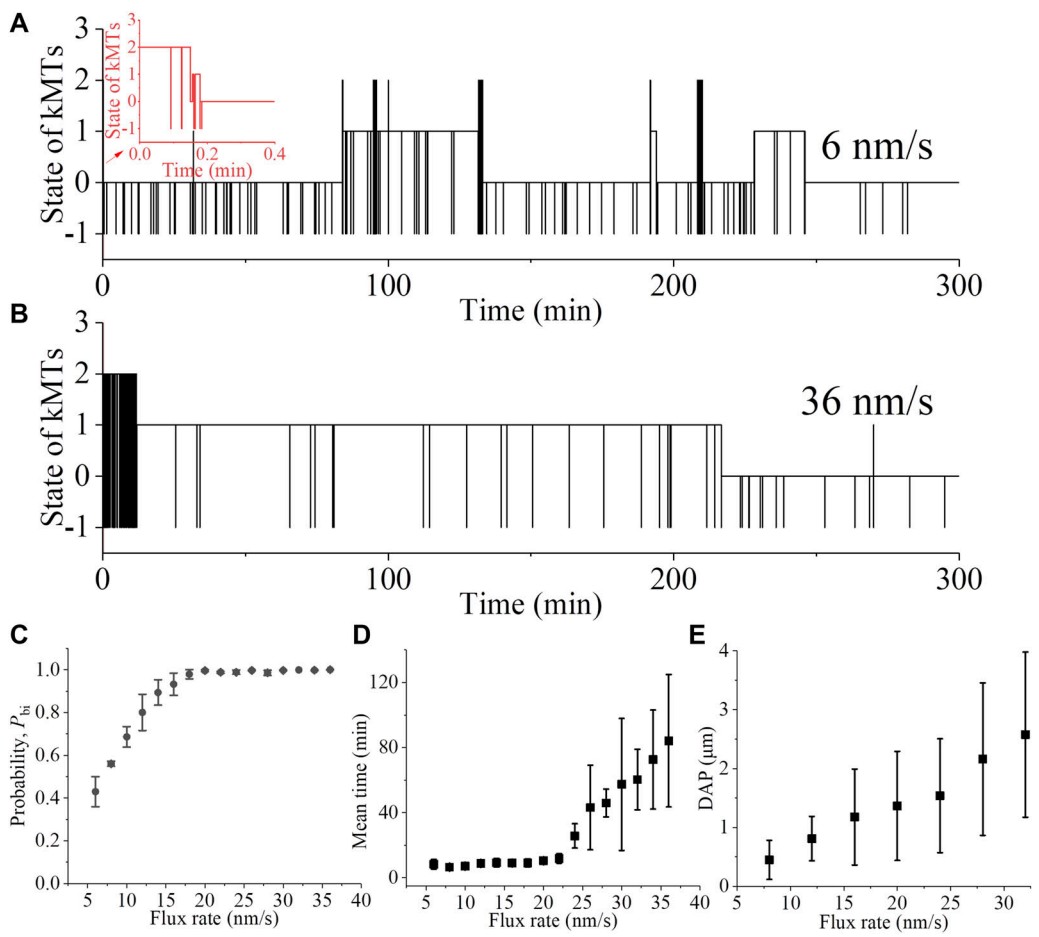

**Figure 8.   Effect of kMT poleward flux on the error correction.**
The simulations begin with the type-II merotelic state in the spindle system composed of *N* = 2 ensembles of MTs (Fig 1C). **(A)** Temporal evolution of the occurrences of the type-I merotelic state, type-II merotelic state, and bi-oriented state at the kMT-flux rate of about 6 nm/s. The inset shows the results in the region between 0 and 0.4 min. **(B)** Temporal evolution of the occurrences of the type-I merotelic state, type-II merotelic state, and bi-oriented state at the kMT-flux rate of about 36 nm/s. In (A, B), when the state with no incorrect attachment (bi-oriented attachments) occurs, it is set as the value equal to "0"; when the state with one incorrect attachment (type-I merotelic attachments) occurs, it is set as the value equal to "1"; when the state with two incorrect attachments (type-II merotelic attachments) occurs, it is set as the value equal to "2"; and the state with the occurrence of kMT detachment is set as the value equal to "-1." **(C)** Probability of correct bi-oriented state, $P_{bi}$, versus kMT-flux rate after the erroneous attachments transitioning to the correct bi-oriented attachments for the first times. Each data point (mean ± SEM) is obtained from six independent 20,000-s trajectories. **(D)** The mean first time for the erroneous attachments to transition to the correct bi-oriented attachments versus kMT-flux rate. For $k_{flux} ≤ 28$ nm/s, each data point (mean ± SEM) represents the average of 20 independent measurements, and for $k_{flux} > 28$ nm/s, each data point (mean ± SEM) represents the average of six independent measurements. **(E)** Relationship between the kMT-flux rate and kinetochore-oscillation amplitude. The oscillation amplitude is characterized by the deviation from the average position (DAP) for the two kinetochores, which is calculated as the absolute value (mean ± SD) of the deviation of the center position of the two kinetochores from the center position of the spindle. In the simulation of (E), for simplicity, the MTs are considered not to be detached and all MTs are taken as in bi-oriented attachments.

spindle pole, making the longer kMT (kMT 1) more likely to experience a pushing force from the kinetochore toward the spindle pole. The low $k_{flux}$ gives a long time period between the arrival of a kinesin-13 motor to the minus end of kMT 1 and the arrival of the kinesin-13 motor to the minus end of kMT 2, giving a long time period for the large difference in the MT lengths between the two kMTs. Thus, the longer kMT 1 can detach with a larger probability. When this longer kMT 1 detaches, the position of its plus end is closer to the right kinetochore, favoring a transition from the state in Fig 1A to that in Fig 1B. Therefore, a low $P_{bi}$ would be expected. To be consistent with this explanation, our simulation results show a higher frequency of the pushing force acting on kMTs at the low flux rate (see Fig S7A in the Supporting Information).

In contrast, the high $k_{flux}$ gives a short time period between the arrival of a kinesin-13 motor to the minus end of one kMT and the arrival of a kinesin-13 motor to the minus end of the other emanated from the same spindle pole, giving a small time-averaged change in the length of one kMT relative to that of the other kMT. In the type-II merotelic state (Fig 1C), if an incorrectly attached kMT detaches from the kinetochore, it would most probably reattach to the kinetochore, maintaining the type-II merotelic state (Fig 1C). If the incorrectly attached kMT detaches from the kinetochore and attaches to the other kinetochore, giving the correctly attached kMT, the pushing force on the new correctly attached kMT would compress the two kinetochores. The new correctly attached kMT would detach rapidly from the kinetochore before it can shorten

sufficiently and thus would most probably attach to the other kinetochore, returning to the type-II merotelic state (Fig 1C). Therefore, a long $\tau$ would be expected. This explanation is consistent with our simulation results showing that the time-averaged change of the interkinetochore distance, $x–x_0$, at the moment when the system has just transitioned to the type-I merotelic attachments has a small value at high $k_{flux}$ (see Fig S7B in the Supporting Information).

Taken together, we show that the kMT poleward flux of a low rate results in a low mitotic fidelity, whereas the kMT flux of a high rate results in the inability of the efficient correction of the initially merotelic attachments before the anaphase onset. Only the flux of an optimum rate that is around 20 nm/s, as measured experimentally in the normal human spindle, can result in both the efficient error correction and the high mitotic fidelity. In other words, the kMT flux of an optimum rate is critical to the efficient error correction. Because the kMT-flux rate is proportional to the kinetochore-oscillation amplitude, we also conclude that the kinetochore oscillation of an optimum amplitude is critical to the error correction.

## Discussion

Here, we study numerically the correction of various erroneous (merotelic, syntelic, and monotelic) attachments in the spindle system with the presence of the MT poleward flux. We show that the erroneous merotelic attachments can be efficiently corrected, with the finally stable spindle having the fidelity (Figs 3, 4, and 5). The syntelic and monotelic attachments can be corrected finally to the bi-oriented attachments (Fig 6B). Moreover, by comparing the results shown in Figs 3, 4, and 5 with those shown in Fig 7 we note that Aurora B activity is indispensable to the highly efficient correction of erroneous attachments, which is consistent with the published experimental results (Sen et al, 2021; Wang et al, 2024).

Previous experimental studies showed that the lagging chromosomes and micronuclei can be rescued by treatments that slowed down the kMT poleward flux (Risteski et al, 2024), implying that the erroneous attachments can be corrected by reducing the rate of the kMT flux. Here, we show that the kMT flux of the high rate can result in the merotelic attachments to be unable to be corrected efficiently (Fig 8D), implying that the initially erroneous merotelic attachments cannot be corrected before the anaphase onset, which is consistent with the previous experimental studies (Risteski et al, 2024). Moreover, we find that the kMT flux of the low rate can result in a low mitotic fidelity (Fig 8C). Because the kMT-flux rate is proportional to the amplitude of the kinetochore oscillation (Fig 8E), the kinetochore oscillation of the small amplitude can result in a low mitotic fidelity, which is consistent with the experimental results observed by Iemura et al (2021). Collectively, we conclude that only the kMT flux of the optimum rate can result in both the efficient error correction and the high mitotic fidelity. This gives an explanation of why the kMT flux was present in the spindle of higher eukaryotes (Steblyanko et al, 2020; Risteski et al, 2022).

For simplicity, we consider that each MT has a long length, which could be applicable to the cells of small sizes. For cells of large sizes, each long "kMT" is composed of a stem kMT, which connects directly to the kinetochore, and many smaller kMTs, with all kMTs being connected with each other via augmin complexes (Kamasaki et al, 2013; Verma & Maresca, 2019) located in the parallel overlap regions, as proposed before (Wang et al, 2025). Similarly, each long "bMT" is composed of a stem bMT, which forms an antiparallel overlap with another stem bMT near the spindle center, and many smaller bMTs, with all bMTs being connected with each other via the augmin complexes located in the parallel overlap regions (Brugues et al, 2012; Wang et al, 2025). The stem kMTs can detach from and reattach to kinetochores, as done in this work. Thus, it is expected that for the cells of large sizes the correction of erroneous attachments of stem kMTs to kinetochores can be realized with the same mechanism as that for the cells of small sizes.

Finally, it should be mentioned that for simplicity of analysis, in the model we have not included the polar ejection force acting on the kinetochore through the chromosome arms (Cane et al, 2013; Li et al, 2016) and spindle pole. It is evident that the effect of the inclusion of the polar ejection force on the spindle dynamics is equivalent to the decrease in the concentration of kinesin-5 motors and thus has no effect on the conclusion presented here. The effect of Aurora A on the error correction is not considered, which will be done in future numerical studies. In addition, for simplicity of analysis, in this work we study the correction of the erroneous attachments using the one-dimensional model, where all MTs are aligned in one direction. Thus, we have not considered the effect of geometric constraints on the attachment of kMTs to kinetochores. The inclusion of the effect of the geometric constraints requires extending the one-dimensional model to the three-dimensional model, which will be done in future numerical studies.

## Materials and Methods

### Monte Carlo simulation method

In the Monte Carlo simulations of the stepping, binding, and unbinding of kinesin-5 and kinesin-13, the binding of NuMA, the MT polymerization and depolymerization activities, and the detachment and reattachment of the kMT to the kinetochore, we take the time step $h = 10^{-3}$ s. We have checked that doubling the time step $h$ does not affect our results.

For each kinesin-5 motor in the antiparallel MT overlap region, we take 4 independent random variables uniformly distributed between 0 and 1, $ran1$, $ran2$, $ran3$, and $ran4$. During each time step $h$, if $ran1 < k_{off}^{(m)} h$, the pair of heads bound to one MT detaches from the MT, where $k_{off}^{(m)}$ is the dissociation rate of one pair of kinesin-5 heads during stepping along one MT (Wang et al, 2025). The equation for $k_{off}^{(m)}$ as functions of the parameters listed in Table S1 was given before (Wang et al, 2025). If $ran2 < k_F^{(m)}h$, the pair of heads bound to one MT takes a forward step, and if $ran3 < k_B^{(m)}h$, the pair of heads bound to one MT takes a backward step, where $k_F^{(m)}$ and $k_B^{(m)}$ are forward and backward stepping rates of one pair of

kinesin-5 heads along one MT, respectively (Wang et al, 2025). The equations for $k_F^{(m)}$ and $k_B^{(m)}$ as functions of the parameters listed in Table S1 were given before (Wang et al, 2025). When one pair of heads is detached from one MT and the other pair of heads of the kinesin-5 motor is bound to another MT in the antiparallel MT overlap zone, if $ran4 < \mu_m h$, the detached pair of heads rebinds to the MT, where $\mu_m$ is the rebinding rate of the detached pair of heads to the MT (see Table S1). For the kinesin-5 motor in the solution binding to one MT, we take 1 independent random variable uniformly distributed between 0 and 1, $ran5$. If $ran5 < k_{on}^{(m)}[K5]h$, one pair of heads of a kinesin-5 motor in the solution binds to an unoccupied tubulin of one MT, where $k_{on}^{(m)}$ is the second-order MT-binding rate, and [K5] is the kinesin-5 concentration (see Table S1).

For the stepping, binding, and unbinding of the kinesin-13, we take 4 independent random variables uniformly distributed between 0 and 1, $ran6$, $ran7$, $ran8$, and $ran9$. During each time step $h$, if $ran6 < k_{diff}^{(K13)} h$, the kinesin-13 bound to MT takes a forward step, and if $ran7 < k_{diff}^{(K13)} h$, the kinesin-13 bound to MT takes a backward step, where $k_{diff}^{(K13)}$ is the forward or backward stepping rate of kinesin-13 along MT because of diffusion (see Table S2). The stepping can only occur when the adjacent tubulin is unoccupied. If $ran8 < k_{off}^{(K13)} h$, the kinesin-13 bound to MT detaches from the MT, where $k_{off}^{(K13)}$ is the detachment rate of kinesin-13 from MT (see Table S2). If $ran9 < k_{on}^{(K13)}[K13] dh$, the kinesin-13 in the solution binds to an unoccupied tubulin in the region that does not form an antiparallel MT overlap, where $k_{on}^{(K13)}$ is the second-order binding rate of kinesin-13 to MT, and [K13] is the kinesin-13 concentration (see Table S2).

For the MT polymerization at the plus end of each MT with the polymerization rate $k_p$, we take one random variable uniformly distributed between 0 and 1, $ran10$. If $ran10 < k_p h$, one tubulin is added to the plus end of the MT, increasing the MT length by $d$. For the bMT and the kMT detached from the kinetochore, the polymerization rate $k_p = v_{p0}/d$ is constant. For the kMT attached to the kinetochore, the polymerization rate $k_p = v_{pol}^{(kMT)}/d$ is dependent on the force on the plus end, as calculated by Equation (1).

For the MT depolymerization at the MT minus end of each MT with the depolymerization rate $k_{dep}$, we take one random variable uniformly distributed between 0 and 1, $ran11$. If a kinesin-13 protein is located at the first tubulin from the minus end and $ran11 < k_{dep}h$, the first tubulin is removed from the minus end of the MT, decreasing the MT length by $d$. If the second tubulin from the minus end is occupied, the kinesin-13 protein detaches from MT after depolymerizing the first tubulin from the minus end. If the second tubulin from the minus end is unoccupied, the kinesin-13 takes a backward step after depolymerizing the first tubulin from the minus end, remaining bound to the new first tubulin from the minus end. The kinesin-13 motor can reside at the minus end for a time $\tau_{end}$ before detachment.

For the NuMA in the solution binding to one bMT, we take 1 independent random variable uniformly distributed between 0 and 1, $ran12$. If $ran12 < k_{on}^{(NuMA)}[NuMA]h$, one MT-binding domain of NuMA in the solution binds to an unoccupied tubulin of bMT, where $k_{on}^{(NuMA)}$ is the second-order binding rate of NuMA to MT, and [NuMA] is the NuMA concentration (see Table S3). For NuMA with one MT-binding domain connected to one bMT, we take one independent random variable uniformly distributed between 0 and 1, $ran13$. During each time step $h$, if $ran13 < \mu_{NuMA} h$, the detached MT-binding domain of NuMA binds to the parallel bMT, where $\mu_{NuMA}$ is the binding rate of one head to MT when another head at the opposite end of the stalk is attached to another parallel MT (see Table S3).

For the kMT detachment from and reattachment to the kinetochore, we take 2 random variables uniformly distributed between 0 and 1, $ran14$, $ran15$. The attached kMT detaches from the kinetochore when $ran14 < k'_{detach} h$ in the presence of Aurora B, or $ran14 < k_{detach}h$ in the absence of Aurora B. For the detached kMT, if $ran15 < k_{attach} h$, the kMT reattaches to the kinetochore.

Note that the activities of kinesin-5 motor, kinesin-13 motor, NuMA protein, MT polymerization and MT depolymerization, and the kMT detachment from and reattachment to the kinetochore are independent of each other.

# Data Availability

The data are available from the corresponding author upon reasonable request.

# Supplementary Information

# Acknowledgements

This work was supported by the Hangzhou Tsientang Education Foundation.

## Author Contributions

Y Wang: data curation, formal analysis, methodology, software, validation, investigation, and writing—original draft, review, and editing.
J Wang: investigation and writing—review and editing.
Y-R Liu: methodology, and writing—review and editing.
P-Y Wang: funding acquisition, methodology, and writing—review and editing.
P Xie: conceptualization, formal analysis, supervision, investigation, methodology, visualization, project administration, and writing—original draft, review, and editing.

## Conflict of Interest Statement

The authors declare that they have no conflict of interest.

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
