## [Reviewer comments · Life Science Alliance]

An optimum rate of microtubule flux for error correction in metaphase spindle

Yao Wang, Jie Wang, Yu-Ru Liu, Peng-Ye Wang and Ping Xie

DOI: <https://doi.org/10.26508/lsa.202503612>

Corresponding author(s): Dr. Ping Xie (Institute of Physics; Chinese Academy of Sciences; University of Chinese Academy of Sciences)

Review Timeline:

Submission Date:	2025-12-24
Editorial Decision:	2026-02-11
Revision Received:	2026-03-13
Editorial Decision:	2026-04-03
Revision Received:	2026-04-10
Accepted:	2026-04-16

Scientific Editor: Tim Fessenden

Transaction Report:

February 11, 2026

Re: Life Science Alliance manuscript #LSA-2025-03612-T

Dr. Ping Xie
Institute of Physics
Laboratory of Soft Matter Physics
Chinese Academy of Sciences, Beijing 100080
Beijing 100190
China

Dear Dr. Xie,

Thank you for submitting your manuscript entitled "An optimum rate of microtubule flux for error correction in metaphase spindle". The manuscript has been evaluated by expert reviewers, whose reports are appended below. Unfortunately, after an assessment of the reviewer feedback, our editorial decision is against publication in Life Science Alliance at this time.

As you will see, reviewers disagreed in their appraisal of this study. Reviewers 1 and 3 remarked on the importance of these findings, and Reviewer 3 expressed strong support with overall minor requests and suggestions to improve this work. However Reviewers 1 and 2 remarked on several faulty assumptions and apparent errors in the computational model. Although your manuscript is intriguing, these issues preclude further consideration of this study in its current form. If you wish to expedite publication of these findings, it may be best to pursue publication at another journal.

Given our interest in the topic, I would be open to re-submission to Life Science Alliance of a significantly revised manuscript that fully addresses the reviewers' main concerns and is subject to further peer review. A suitably revised manuscript must address points 1-4 and point 7 raised by Reviewer 1. I note that point 1 concerns changes to equations that underlie this model, so it is possible that resolving this concern will alter key conclusions. A revised manuscript should further either justify or improve key assumptions made by this model noted by Reviewer 2.

If you would like to resubmit this work to Life Science Alliance, you may submit an appeal directly through our manuscript submission system. The appeal request must include a cover letter and a rebuttal with point-by-point responses to the reviewer comments. A fully revised manuscript is not required for the appeal request. Should you decide to pursue an appeal of this decision, I suggest we discuss the requested revisions in more detail via email or phone/videoconferencing. Please let me know which option you prefer.

Regardless of how you choose to proceed, we hope that the comments below will prove constructive as your work progresses.

Thank you for thinking of Life Science Alliance as an appropriate venue to publish your work.

Sincerely,

Reviewer #1 (Comments to the Authors (Required)):

In this theoretical study, Wang and colleagues investigate the evolution of kinetochore-microtubule attachments. To explore the conditions under which erroneous attachments are corrected, the authors extend their previous model (Wang et al., 2025) by incorporating several physical processes reported in experimental studies. In the proposed framework, kinetochore detachment depends on the tensile force between the kinetochore and microtubule as well as on the inter-kinetochore distance, whereas attachment depends on the predicted future tensile force. The authors use this model to interpret biological observations related to Aurora B activity and poleward flux reported in the literature.

Although the biological question addressed is interesting and relevant, the manuscript in its current form requires major revision. In particular, substantial changes to the model formulation and a clearer presentation and substantiation of the results are needed. Below, I outline my major and minor concerns.

Major comments

(1) Incorrect formulation of key equations. Two equations central to the model appear to be incorrect:

- Equation (4) defines the probability of detachment, P_{detach} . As written, this quantity can exceed 1, which contradicts the basic definition of a probability.
- Equation (5) assumes that detachment depends on the future tensile force. This assumption is inconsistent with the stated model framework in which each microtubule is attached to an independent spring. A physically consistent description would treat the free spring as being subject to thermal fluctuations, with its extension determined by a Boltzmann distribution. A closely related situation has been analyzed in the context of spindle oscillations (Grill et al., Phys. Rev. Lett. 2005). Because these equations are fundamental to the model, the authors should revise the formulation and repeat the calculations using a physically consistent set of equations.

(2) Inadequate description of Aurora B function. The role of Aurora B is not sufficiently defined. Based on Equations (3) and (4), its effect appears to be limited to enhancing microtubule detachment at small inter-kinetochore distances. This is likely an oversimplification of its biological function. Several studies indicate that Aurora B localizes predominantly to the inner centromere-a chromatin region positioned between the inter-kinetochore and inter-sister chromatid axes (Hindriksen et al., Front. Cell Dev. Biol. 2017), or to multiple discrete kinetochore and centromere regions (Broad et al., J. Cell Biol. 2020, e201905144). The authors should revise their equations to better reflect the known spatial localization and regulatory role of Aurora B.

(3) Oversimplified microtubule geometry. The model considers only two microtubules extending toward a single kinetochore. This is far from a realistic spindle configuration, where many microtubules emanate from each spindle pole. This simplification cannot be applied when considering the attachment of newly nucleated microtubules because it may lead to an unequal number of microtubules from opposite poles for at location of asymmetrically positioned chromosomes (see Sigmund et al., PRX Life 2024). The authors should justify whether and under what conditions a two-microtubule model can adequately represent realistic spindle dynamics.

(4) Unsupported statements in the Results section. Several statements in the manuscript are not supported by the presented results. For example, the sentence "Taken together, we show that with the catch-slip-bonding interaction of the kMT with kinetochore (Fig. S2), due to the effect of Aurora B activity, only the attachment in the spindle having the correct bi-oriented attachments has the maximum lifetime whereas the attachment in the spindle having erroneous attachments has the shorter lifetime" is not supported by any figure. Attachment lifetimes are not shown, and results with and without Aurora B activity are not compared. The manuscript should be substantially revised to ensure that all claims are clearly and quantitatively supported by the data.

(5) Missing time-resolved analysis of key events. Information on the time course of critical events is lacking. For instance, it would be highly informative to show how the number of unattached kinetochores that achieve biorientation evolves over time, as well as how the number of erroneous attachments changes during the simulation. Such analyses would greatly improve the interpretability of the model.

(6) Biological relevance of reported error rates. In Figure 4, the authors identify conditions under which the number of erroneous attachments is below 4%. However, this still corresponds to a large absolute number of errors. For a human spindle with 46 chromosomes, a simple estimate gives $46 \times 4\% / 2 \approx 0.92$ erroneous attachments per cell. The authors should discuss the relevance of their model in the context of realistic error rates, given that normal cells in missegregate chromosomes approximately 0.5-5 times per 100 cell divisions (Klaasen & Kops, Cells 2022).

(7) Poor clarity and lack of statistical analysis in Figures 5-8. Figures 5-8 show the evolution of erroneous attachments, but these plots are difficult to interpret. They contain multiple spikes appearing in an apparently random order, and no systematic statistical analysis is provided. In addition, left and right spindle sides are not shown separately, and in Figure 6a there is an unexplained black stripe between -1 and 0 on the y-axis. These issues prevent meaningful scientific conclusions from being drawn. Please redesign these figures to make your scientific conclusions clearer.

(8) Ambiguity in Figure 6. Figure 6 shows time courses for a single event, but it is unclear whether panels a-d correspond to the same simulation run. Ideally, these panels should be derived from the same run and shown in one column so that correlations between kinetochore movement and attachment type can be directly assessed.

Minor comment

- (1) Please use the term bridging microtubule instead of ipMT, as it is more accurate.

Reviewer #2 (Comments to the Authors (Required)):

This manuscript by Wang et al. presents a computational modeling effort to describe the process of resolving errors in

kinetochore-microtubule attachment. Prior work by the lab established a framework for this paper that incorporates microtubule polymerization rates; kinetochore-derived forces that pull on kinetochore-microtubule plus ends; and a role for Aurora B in the error correction process. This study adds polewards microtubule flux as a component of the error correction machinery and is motivated by work recently published by the Tolic laboratory (Risteski et al. (2024)). Using Monte-Carlo simulations of their mechanochemical spindle model, the authors investigate how all known errors in kinetochore-microtubule attachment (merotelic, syntelic, monotelic) are corrected prior to anaphase. The study demonstrates three central findings. First, Aurora B kinase for efficient error correction. Second, the authors show that all classes of attachment errors can be corrected when Aurora B activity and MT flux are present. Merotelic attachments resolve efficiently through biased detachment-reattachment dynamics; syntelic and monotelic attachments are either trapped near spindle poles (when no opposing MTs can grow) or ultimately corrected once MTs from the opposite pole are allowed to polymerize. Third, the model reveals that error correction requires an optimal MT poleward flux rate. Low flux (and correspondingly small kinetochore oscillations) leads to frequent loss of correct attachments and reduces mitotic fidelity, while excessively high flux prevents efficient resolution of merotelic attachments. In summary, only an intermediate flux rate (which mimics flux rate in human cells (~20 nm/s)), simultaneously yields rapid error correction.

Overall, this study provides a plausible mechanistic explanation for why MT poleward flux exists in higher eukaryotes and why its magnitude is tightly regulated. It links Aurora B-mediated tension sensing, MT dynamics, and chromosome oscillations into a coherent framework that accounts for both high fidelity and robustness of chromosome segregation. My main issue is that this study does not provide any new insight into error correction mechanism. All of the central findings have already been established through experiment. Moreover, the authors make a few assumptions that are not appropriately justified. For example, values for kinetochore-attachment strength and microtubule polymerization/depolymerization rates are taken from experiments conducted using yeast kinetochores. Second, the authors make the assumption that minus ends are readily available for kinesin-13-mediated depolymerization. Many minus ends in the kinetochore are likely protected from kinesin-13 by the gamma-tubulin ring complex; decapping - a process not well-understood in living cells - is required for MCAK or KIF2A to target minus ends (and thereby drive flux). Third, kinetochore oscillations - which the authors argue to be central for error correction - are not incorporated into the model. Aside from these major weaknesses, there are many grammatical and spelling errors in the manuscript. Collectively, my disposition is that this paper is not appropriate for publication in Life Science Alliance in its current form.

Reviewer #3 (Comments to the Authors (Required)):

The manuscript "An optimum rate of microtubule flux for error correction in metaphase spindle" by Wang et al. explores the mechanisms on error corrections for kinetochore-microtubules (kMT) attachments in metaphase spindle. Particularly, they study numerically the error correction corresponding to merotelic, syntelic and monotelic attachments in the presence of MT poleward flux. Their starting point is a previously proposed model by the authors (Wang et al., 2025) where they studied the stability of the spindle during metaphase and how the presence of MT poleward flux regulates the spindle length. In the present study they dig deeper into the intrinsic role of kMT dynamics to correct erroneous attachments efficiently. Additionally, they here incorporate the analysis of a model that has been suggested to play a role in the error-correction mechanism: the effect of Aurora B kinase activity in the regulation of MT dynamics and the stability for kMT attachments. They found that Aurora B kinase is crucial for efficient correction of erroneous attachments. Moreover, author examined the role of low and high MT-flux rates, that is proportional to low or high amplitude of kinetochore oscillations. Their results showed that low rates of MT-flux lead to low mitotic fidelity, while higher MT-flux rates result in the inability to efficiently correct erroneous attachments. They conclude that only an optimum MT-flux rate of 20 nm/s can result in efficient error correction and in high mitotic fidelity.

Overall, this is a remarkable study showing that key mechanisms that participates for a precise and accurate division, and the deviation from the correct normal conditions, can be numerically modeled taking into account the experimental observed evidence.

This reviewer would appreciate clarification from the authors on the following points.

Minor comments

1. An important point on this study is the probability for recovery the correct attachments of kMT, and this probability is calculated in the frame of a time duration for the simulations that, for instance, in Line 271 is established to be 333 seconds, and varies for other conditions throughout the manuscript. This process for erroneous correction attachments takes place in real cells within a time window that is lasting shorter. How do the authors reconcile the metaphase-to-cytokinesis duration observed in real cells with the attachment-recovery time predicted by their simulations for the probability they calculated? The manuscript would benefit from additional contextualization from relevant literature on characteristic real metaphase-to-cytokinesis typical duration.

2. The authors present throughout the manuscript the possibility of kMT attachment-detachment and correct attachment states, for instance: "After the first transitioning to the bi-oriented state, the spindle system maintains almost at the bi-oriented state, although the system can occasionally transition back to type-I merotelic state and kMT detachment can occur frequently. If the type-I merotelic state or kMT detachment occurs, the system can return rapidly to the bi-oriented state." How often do these attachments/detachments occur in real cells under experimental conditions?

3. In page 13, the authors study the effect of the number of ensembles of MT by analyzing the case of $N=3$, resulting that "It is seen that in in about 2.6 min (156 s) the erroneous attachments transition to the correct bi-oriented attachments for the first times". However, previously they claim than in the case $N=2$ "It is seen that in about 5.1 min (306 s) the erroneous attachments transition to the correct bi-oriented attachments for the first times". And later they conclude: "These results for $N = 3$ ensembles of MTs (Fig. 5) are similar to those for $N = 2$ ensembles of MTs (Fig. 3), indicating that our conclusion is independent of N ." However, one could deduce that in the case of $N=3$ the erroneous attachments are corrected in a shorter period of time leading to a more accurate system. Thus, the number of MT is, in fact, ensuring higher mitotic fidelity. Then, the statement of the authors is not fully supported. Have the authors explored higher number of MT ensembles? If yes, what are the times for the first erroneous attachments transition?

4. Related with question 1, in Line 364 "Taken together, we show that the merotelic attachments can be corrected efficiently, with the finally stable kinetochore-kMT attachments having the fidelity of at least 0.96." However, to my understanding the key point (for a cell to accomplish correct stable attachments) would rather be the lasting time to reach this fidelity attachment, since the metaphase-to-cytokinesis duration is preset for cells.

5. The authors have shown how the activity of Aurora B is essential to the efficient correction of erroneous attachments. Have the authors tried to saturate the system by increasing the Aurora B -kinase presence? Have they thought or explore how is this affecting the attachment/detachment and the MT-flux rates?

Responses to Reviewer #1

We are very grateful to the reviewer for his/her valuable comments on our manuscript. These comments have contributed a lot to improve the quality of our manuscript. According to the suggestions and comments, we have revised the manuscript.

Reviewer #1 (Comments to the Authors (Required)):

In this theoretical study, Wang and colleagues investigate the evolution of kinetochore-microtubule attachments. To explore the conditions under which erroneous attachments are corrected, the authors extend their previous model (Wang et al., 2025) by incorporating several physical processes reported in experimental studies. In the proposed framework, kinetochore detachment depends on the tensile force between the kinetochore and microtubule as well as on the inter-kinetochore distance, whereas attachment depends on the predicted future tensile force. The authors use this model to interpret biological observations related to Aurora B activity and poleward flux reported in the literature.

Although the biological question addressed is interesting and relevant, the manuscript in its current form requires major revision. In particular, substantial changes to the model formulation and a clearer presentation and substantiation of the results are needed. Below, I outline my major and minor concerns.

Major comments

(1) Incorrect formulation of key equations. Two equations central to the model appear to be incorrect:

- Equation (4) defines the probability of detachment, P_{detach} . As written, this quantity can exceed 1, which contradicts the basic definition of a probability.

Response: We apologize that the symbol P_{detach} [defined in Eq. (4)] used in our previous version causes misleading. Thus, in the revised version, we use X_{detach} instead of P_{detach} . As defined (see red words in page 7), X_{detach} is the times of the detachment rate enhanced by the effect of Aurora B activity relative to that in the absence of Aurora B, rather than the probability of detachment. Since X_{detach} is the times, its quantity can exceed 1.

- Equation (5) assumes that detachment depends on the future tensile force. This assumption is inconsistent with the stated model framework in which each microtubule is attached to an independent spring. A physically consistent description would treat the free spring as being subject to thermal fluctuations, with its extension determined by a Boltzmann distribution. A closely related situation has been analyzed in the context of spindle oscillations (Grill et al., Phys. Rev. Lett. 2005). Because these equations are fundamental to the model, the authors should revise the formulation and repeat the calculations using a physically consistent set of equations.

Response: In the previous version, the tensile force in Eq. (5) represents the force on each microtubule at the moment when it attaches to kinetochore, which is dependent only on the

corresponding spring and the distance between the plus end of the detached MT and kinetochore. In the revised version, Eq. (5) is revised, where we use the spring elastic coefficient times the distance of the plus end of the unattached microtubule to a kinetochore instead of the force. It is noted that Eq. (5) is similar to that used by Grill et al. (i.e., Eq. (11) in Grill et al.) for the microtubule detachment rate (see red words in page 7).

(2) Inadequate description of Aurora B function. The role of Aurora B is not sufficiently defined. Based on Equations (3) and (4), its effect appears to be limited to enhancing microtubule detachment at small inter-kinetochore distances. This is likely an oversimplification of its biological function. Several studies indicate that Aurora B localizes predominantly to the inner centromere-a chromatin region positioned between the inter-kinetochore and inter-sister chromatid axes (Hindriksen et al., *Front. Cell Dev. Biol.* 2017), or to multiple discrete kinetochore and centromere regions (Broad et al., *J. Cell Biol.* 2020, e201905144). The authors should revise their equations to better reflect the known spatial localization and regulatory role of Aurora B.

Response: In this work, to be consistent with the experimental data showing that Aurora B localizes predominantly to the inner centromere region (Hindriksen et al., 2017; Yamamoto, 2021), for simplicity, we only consider the effect of Aurora B localizing to the inner centromere region and do not consider the effect of Aurora B localizing to other regions (Broad et al., 2020). In the revised version, some words were added to emphasize this point (see blue words in page 7).

(3) Oversimplified microtubule geometry. The model considers only two microtubules extending toward a single kinetochore. This is far from a realistic spindle configuration, where many microtubules emanate from each spindle pole. This simplification cannot be applied when considering the attachment of newly nucleated microtubules because it may lead to an unequal number of microtubules from opposite poles for at location of asymmetrically positioned chromosomes (see Sigmund et al., *PRX Life* 2024). The authors should justify whether and under what conditions a two-microtubule model can adequately represent realistic spindle dynamics.

Response: For the sake of saving computation time, we used two microtubules extending toward a single kinetochore in our simulation, which are smaller than those in a realistic spindle configuration. Under our current simulation capacity, it is unable to make the simulation with the realistic number of microtubules in the spindle configuration, because this will involve too many motor proteins such as kinesin-5/Eg5, kinesin-13, NuMA, etc., and the simulation of the activities of those proteins takes too long time. To check if our results are valid using two microtubules, we make another simulation using three microtubules (see Fig. 5) and we note that the results using three microtubules are similar to those using two microtubules. Moreover, based on the mechanism of error correction in our model (see red words in pages 12 and 13), it is noted that our conclusion should be applicable to any number of microtubules. Thus, our conclusions deduced from the simulation using two or three microtubules are applicable to the realistic number of microtubules. In our studies we also consider the attachment of newly nucleated microtubules (see the results presented in Fig. 6b). In addition, the paper of Sigmund et al., *PRX Life* 2024 is cited in the revised manuscript.

(4) Unsupported statements in the Results section. Several statements in the manuscript are not supported by the presented results. For example, the sentence "Taken together, we show that with the catch-slip-bonding interaction of the kMT with kinetochore (Fig. S2), due to the effect of Aurora B activity, only the attachment in the spindle having the correct bi-oriented attachments has the maximum lifetime whereas the attachment in the spindle having erroneous attachments has the shorter lifetime" is not supported by any figure. Attachment lifetimes are not shown, and results with and without Aurora B activity are not compared. The manuscript should be substantially revised to ensure that all claims are clearly and quantitatively supported by the data.

Response: The statement "Taken together, we show that with the catch-slip-bonding interaction of the kMT with kinetochore (Fig. S2), due to the effect of Aurora B activity, only the attachment in the spindle having the correct bi-oriented attachments has the maximum lifetime whereas the attachment in the spindle having erroneous attachments has the shorter lifetime" is deduced from the results of Fig. 2a, where it is shown that only for the correct bi-oriented attachments the pulling force on each kMT is 5 pN and the inter-kinetochore distance has the largest value. In the revised version, we added a figure (Fig. S5) showing the attachment lifetime for various attachments. The results of Fig. S5 support our above statement. See blue words in page 10.

(5) Missing time-resolved analysis of key events. Information on the time course of critical events is lacking. For instance, it would be highly informative to show how the number of unattached kinetochores that achieve biorientation evolves over time, as well as how the number of erroneous attachments changes during the simulation. Such analyses would greatly improve the interpretability of the model.

Response: In Figs. 3a, 5, 6a (left panel), 7a, 8a and 8b we show how the number of erroneous attachments changes and that of kMT detachment change during the simulation. For clarity, we added Fig. 3b, where we only show how the number of erroneous attachments changes, namely we exclude the occurrence of kMT detachment. See blue words in page 12.

(6) Biological relevance of reported error rates. In Figure 4, the authors identify conditions under which the number of erroneous attachments is below 4%. However, this still corresponds to a large absolute number of errors. For a human spindle with 46 chromosomes, a simple estimate gives $46 \times 4\% / 2 \approx 0.92$ erroneous attachments per cell. The authors should discuss the relevance of their model in the context of realistic error rates, given that normal cells in missegregate chromosomes approximately 0.5-5 times per 100 cell divisions (Klaasen & Kops, Cells 2022).

Response: In our previous version, the fidelity, which is characterized by P_{bi} , was calculated as the time period of the bi-oriented attachments divided by total time period (with the inclusion of the time period of the temporary kMT detachment). Considering that the detached kMT can reattach rapidly (with a time of about 0.044 s, see caption of Fig. 3), the temporary kMT detachment cannot be

considered as the erroneous state. Thus, the calculation of P_{bi} should exclude the time period of the temporary kMT detachment. In other words, P_{bi} should be calculated as the time period of the bi-oriented attachments divided by total time period of either bi-oriented or erroneous attachments. Therefore, in the revised version we re-calculate P_{bi} . We discuss the relevance of this new P_{bi} in the context of realistic error rates and compare with approximately 0.5 – 5 times of missegregate chromosomes per 100 cell divisions in normal cells (Klaasen and Kops, Cells 2022). See blue words in pages 11, 15 and 16.

(7) Poor clarity and lack of statistical analysis in Figures 5-8. Figures 5-8 show the evolution of erroneous attachments, but these plots are difficult to interpret. They contain multiple spikes appearing in an apparently random order, and no systematic statistical analysis is provided. In addition, left and right spindle sides are not shown separately, and in Figure 6a there is an unexplained black stripe between -1 and 0 on the y-axis. These issues prevent meaningful scientific conclusions from being drawn. Please redesign these figures to make your scientific conclusions clearer.

Response: In Figures 3 and 5-8 we show the evolution of corrected bi-oriented and erroneous attachments. Since for both monotelic and syntelic attachments the change of the inter-kinetochore distance, $x - x_0$, has a mean value of zero (Fig. 2), the detachment of a kMT from and reattachment of the detached kMT to a kinetochore can occur frequently. Thus, in the long simulation time of 330 min (Fig. 6a) it looks like a black stripe between -1 and 0 on the y-axis. For clarity, in Fig. 6a we added an inset (the results in the narrow range between 69.85 min and 70.15 min) showing clearly the occurrence of kMT detachment.

(8) Ambiguity in Figure 6. Figure 6 shows time courses for a single event, but it is unclear whether panels a-d correspond to the same simulation run. Ideally, these panels should be derived from the same run and shown in one column so that correlations between kinetochore movement and attachment type can be directly assessed.

Response: We added some words (see red words in the caption of Fig. 6, page 17) to emphasize that both the left and right panels in (a) correspond to the same simulation run while both the left and right panels in (b) correspond to the same simulation run.

Minor comment

(1) Please use the term bridging microtubule instead of ipMT, as it is more accurate.

Response: In the revised version we use bridging microtubule (bMT) instead of iMT.

Responses to Reviewer #2

We are very grateful to the reviewer for his/her valuable comments on our manuscript. These comments have contributed a lot to improve the quality of our manuscript. According to the suggestions and comments, we have revised the manuscript.

Reviewer #2 (Comments to the Authors (Required)):

This manuscript by Wang et al. presents a computational modeling effort to describe the process of resolving errors in kinetochore-microtubule attachment. Prior work by the lab established a framework for this paper that incorporates microtubule polymerization rates; kinetochore-derived forces that pull on kinetochore-microtubule plus ends; and a role for Aurora B in the error correction process. This study adds polewards microtubule flux as a component of the error correction machinery and is motivated by work recently published by the Tolic laboratory (Risteski et al. (2024)). Using Monte-Carlo simulations of their mechanochemical spindle model, the authors investigate how all known errors in kinetochore-microtubule attachment (merotelic, syntelic, monotelic) are corrected prior to anaphase. The study demonstrates three central findings. First, Aurora B kinase for efficient error correction. Second, the authors show that all classes of attachment errors can be corrected when Aurora B activity and MT flux are present. Merotelic attachments resolve efficiently through biased detachment-reattachment dynamics; syntelic and monotelic attachments are either trapped near spindle poles (when no opposing MTs can grow) or ultimately corrected once MTs from the opposite pole are allowed to polymerize. Third, the model reveals that error correction requires an optimal MT poleward flux rate. Low flux (and correspondingly small kinetochore oscillations) leads to frequent loss of correct attachments and reduces mitotic fidelity, while excessively high flux prevents efficient resolution of merotelic attachments. In summary, only an intermediate flux rate (which mimics flux rate in human cells (~20 nm/s)), simultaneously yields rapid error correction.

Overall, this study provides a plausible mechanistic explanation for why MT poleward flux exists in higher eukaryotes and why its magnitude is tightly regulated. It links Aurora B-mediated tension sensing, MT dynamics, and chromosome oscillations into a coherent framework that accounts for both high fidelity and robustness of chromosome segregation. My main issue is that this study does not provide any new insight into error correction mechanism. All of the central findings have already been established through experiment. Moreover, the authors make a few assumptions that are not appropriately justified. For example, values for kinetochore-attachment strength and microtubule polymerization/depolymerization rates are taken from experiments conducted using yeast kinetochores.

Response: The error correction mechanisms are explained in the manuscript. For example, see red words in pages 12 and 13 for the explanation of the mechanism of efficient correction of the merotelic errors; see red words in page 19 for the explanation of the mechanism of low efficient correction of merotelic error in the absence of the Aurora B activity; see red words in page 23 for the

explanation of the mechanism of the optimum rate of microtubule flux for error correction.

The available experimental results showed that the lifetimes of kinetochore-attachment in the spindle of the higher eukaryote [Kabeche and Compton, *Nature* 502, 110 (2013)] is similar to that in yeast spindle [Akiyoshi et al., *Nature* 468, 576-579 (2010)]. Since the force dependence of kinetochore-attachment lifetime in the yeast spindle is available [Akiyoshi et al., *Nature* 468, 576-579 (2010)] whereas in the spindle of the higher eukaryote is unavailable, for the simulation, we take the force dependence of kinetochore-attachment strength in the spindle of the higher eukaryote having the same form as that in the yeast spindle, similar to the discussion in the literature [Barisic and Rajendraprasad, *BioEssays* 2100079 (2021)]. The microtubule polymerization rate at the plus end is consistent with the flux rate for the human spindle. In the revised version, we added some words on these points (see blue words in page 6).

Second, the authors make the assumption that minus ends are readily available for kinesin-13-mediated depolymerization. Many minus ends in the kinetochore are likely protected from kinesin-13 by the gamma-tubulin ring complex; decapping - a process not well-understood in living cells - is required for MCAK or KIF2A to target minus ends (and thereby drive flux).

Response: For simplicity, we assume that the microtubule minus end is depolymerized with a rate k_{dep} (defined in Table S2). As stated by the reviewer, in reality at the minus end the KIF2A motor together with the pole-localized MT severases uncap the γ -tubulin ring complex and then depolymerize microtubule (we added some words to emphasize this point. See red words in page 5). Considering this more complex process, in our model this implies that the rate k_{dep} denotes the rate of the process including both the γ -tubulin ring complex uncapping and microtubule depolymerization.

Alternatively, if after the uncapping the γ -tubulin ring complex is unable to re-cap the minus end within the short time before the next KIF2A reaching the new minus end, k_{dep} denotes the depolymerization rate of the KIF2A motor.

Third, kinetochore oscillations - which the authors argue to be central for error correction - are not incorporated into the model. Aside from these major weaknesses, there are many grammatical and spelling errors in the manuscript. Collectively, my disposition is that this paper is not appropriate for publication in Life Science Alliance in its current form.

Response: As shown in our previous work (Wang et al., 2025, *IScience* 28: 113506) (see also Fig. S3), although the kinetochore oscillations are not incorporated explicitly into the model, our model can automatically result in the slow and large-amplitude kinetochore oscillations. The origin of the oscillations is due to the stochastic minus-ended microtubule depolymerization by KIF2A motors. Because the analysis of the origin of the oscillations is complicated, we will present the detailed analysis in another work. In the revised version, we added some words to mention this point (see red words in page 3).

Responses to Reviewer #3

We are very grateful to the reviewer for his/her valuable comments on our manuscript. These comments have contributed a lot to improve the quality of our manuscript. According to the suggestions and comments, we have revised the manuscript.

Reviewer #3 (Comments to the Authors (Required)):

The manuscript "An optimum rate of microtubule flux for error correction in metaphase spindle" by Wang et al. explores the mechanisms on error corrections for kinetochore-microtubules (kMT) attachments in metaphase spindle. Particularly, they study numerically the error correction corresponding to merotelic, syntelic and monotelic attachments in the presence of MT poleward flux. Their starting point is a previously proposed model by the authors (Wang et al., 2025) where they studied the stability of the spindle during metaphase and how the presence of MT poleward flux regulates the spindle length. In the present study they dig deeper into the intrinsic role of kMT dynamics to correct erroneous attachments efficiently. Additionally, they here incorporate the analysis of a model that has been suggested to play a role in the error-correction mechanism: the effect of Aurora B kinase activity in the regulation of MT dynamics and the stability for kMT attachments. They found that Aurora B kinase is crucial for efficient correction of erroneous attachments. Moreover, author examined the role of low and high MT-flux rates, that is proportional to low or high amplitude of kinetochore oscillations. Their results showed that low rates of MT-flux lead to low mitotic fidelity, while higher MT-flux rates result in the inability to efficiently correct erroneous attachments. They conclude that only an optimum MT-flux rate of 20 nm/s can result in efficient error correction and in high mitotic fidelity.

Overall, this is a remarkable study showing that key mechanisms that participates for a precise and accurate division, and the deviation from the correct normal conditions, can be numerically modeled taking into account the experimental observed evidence.

This reviewer would appreciate clarification from the authors on the following points.

Minor comments

1. An important point on this study is the probability for recovery the correct attachments of kMT, and this probability is calculated in the frame of a time duration for the simulations that, for instance, in Line 271 is established to be 333 seconds, and varies for other conditions throughout the manuscript. This process for erroneous correction attachments takes place in real cells within a time window that is lasting shorter. How do the authors reconcile the metaphase-to-cytokinesis duration observed in real cells with the attachment-recovery time predicted by their simulations for the probability they calculated? The manuscript would benefit from additional contextualization from relevant literature on characteristic real metaphase-to-cytokinesis typical duration.

Response: Since after the spindle system reaching the bi-oriented steady state it can occasionally

transition back to the erroneous state and then return rapidly back to the bi-oriented state (see, e.g., Fig. 3), the steady state is not a stable state but a dynamic equilibrium state. For the calculation of the probability of the spindle in the correct bi-oriented attachments after the spindle reaching the steady state, we use a long simulation time of 333 min (20000 s) to make the calculation. Note that the longer simulation time used in the calculation the more preciseness of the calculated probability is. This long simulation time of 333 min (20000 s) does not reflect the metaphase-to-cytokinesis duration in rear cells. The probability represents the probability of the system in the bi-oriented state at the moment when the transition from the metaphase to anaphase phase occurs. In the revised version, we added some words to make this more clear (see blue words in pages 11, 15 and 16).

It is noted here that in the previous version, the probability P_{bi} was calculated as the time period of the bi-oriented attachments divided by total time period (with the inclusion of the time period of the temporary kMT detachment). Considering that the detached kMT can reattach rapidly (with a time of about 0.044 s, see caption of Fig. 3), the temporary kMT detachment cannot be considered as the erroneous state. Thus, the calculation of P_{bi} should exclude the time period of the temporary kMT detachment. In other words, P_{bi} should be calculated as the time period of the bi-oriented attachments divided by total time period of either bi-oriented or erroneous attachments. Therefore, in the revised version we re-calculate P_{bi} .

2. The authors present throughout the manuscript the possibility of kMT attachment-detachment and correct attachment states, for instance: "After the first transitioning to the bi-oriented state, the spindle system maintains almost at the bi-oriented state, although the system can occasionally transition back to type-I merotelic state and kMT detachment can occur frequently. If the type-I merotelic state or kMT detachment occurs, the system can return rapidly to the bi-oriented state." How often do these attachments/detachments occur in real cells under experimental conditions?

Response: Since our results show that after the spindle reaching the steady state, after the detachment of a kMT it can re-attach rapidly (with a mean time of only about 0.044 s, see caption of Fig. 3). Due to the too short duration time, the attachments/detachments are unable to be detected in real cells under experimental conditions.

3. In page 13, the authors study the effect of the number of ensembles of MT by analyzing the case of $N=3$, resulting that "It is seen that in in about 2.6 min (156 s) the erroneous attachments transition to the correct bi-oriented attachments for the first times". However, previously they claim than in the case $N=2$ "It is seen that in about 5.1 min (306 s) the erroneous attachments transition to the correct bi-oriented attachments for the first times". And later they conclude: "These results for $N = 3$ ensembles of MTs (Fig. 5) are similar to those for $N = 2$ ensembles of MTs (Fig. 3), indicating that our conclusion is independent of N ." However, one could deduce that in the case of $N=3$ the erroneous attachments are corrected in a shorter period of time leading to a more accurate system. Thus, the number of MT is, in fact, ensuring higher mitotic fidelity. Then, the statement of the authors is not fully supported. Have the authors explored higher number of MT ensembles? If yes, what are the times for the first erroneous attachments transition?

Response: In Fig. 3 we show one simulation run for N=2 and in Fig. 5 we show one simulation run for N=3. Due to the stochasticity, one simulation run can give the time different from another simulation run. In the revised version, we added another simulation run for N=2 in Fig. S6, where it is seen that in about 0.9 min (54 s) the erroneous attachments transition to the correct bi-oriented attachments for the first times, which is different from about 5.1 min (306 s) shown in Fig. 3 for N=2 and is smaller than about 2.6 min (156 s) in Fig. 5 for N=3. Considering the stochasticity, it is noted that results for N = 3 (Fig. 5) are similar to those for N = 2 (Figs. 3 and S6). Some words were added to mention this point (see red words in pages 10 and 15).

4. Related with question 1, in Line 364 "Taken together, we show that the merotelic attachments can be corrected efficiently, with the finally stable kinetochore-kMT attachments having the fidelity of at least 0.96." However, to my understanding the key point (for a cell to accomplish correct stable attachments) would rather be the lasting time to reach this fidelity attachment, since the metaphase-to-cytokinesis duration is preset for cells.

Response: It is true that the metaphase-to-cytokinesis duration is preset for cells. Since after reaching the bi-oriented steady state the spindle system can occasionally transition back to the erroneous state and then return rapidly back to the bi-oriented state (see, e.g., Fig. 3), the finally steady state is not a stable state but a dynamic equilibrium state. Thus, the fidelity defined here represents the probability of the system in the bi-oriented state after reaching the dynamic steady state. In other words, the fidelity represents the probability of the system in the bi-oriented state at the moment when the transition from the metaphase to anaphase phase occurs. In the revised version, we added some words to make this more clear (see blue words in pages 11, 15 and 16).

5. The authors have shown how the activity of Aurora B is essential to the efficient correction of erroneous attachments. Have the authors tried to saturate the system by increasing the Aurora B -kinase presence? Have they thought or explore how is this affecting the attachment/detachment and the MT-flux rates?

Response: The results with Aurora B activity are given in Figs. 3 – 5 and described in Section entitled '*Correction of merotelic attachments*' (pages 10 – 16), showing that the fidelity is high and the time to reach the bio-oriented attachments is short. B comparison, the results without Aurora B activity are given in Fig. 7 and described in Section entitled '*Aurora B activity is essential for error correction*' (pages 19-20), showing that the time to reach the bio-oriented attachments is long. In our model, the effect of Aurora B on the detachment is reflected in Eqs. (3) and (4) and Aurora B has no effect on the attachment.

April 3, 2026

RE: Life Science Alliance Manuscript #LSA-2025-03612-TR-A

Dr. Ping Xie
Institute of Physics
Laboratory of Soft Matter Physics
Chinese Academy of Sciences, Beijing 100080
Beijing 100190
China

Dear Dr. Xie,

Thank you for submitting your revised manuscript entitled "An optimum rate of microtubule flux for error correction in metaphase spindle". We returned this manuscript to Reviewers 1 and 2 for evaluation and their comments are below. As you will see, the reviewers disagree on the whether the revised work is now suitable for publication. In accordance with the concerns raised by Reviewer 2, certain assumptions made in this work should be directly stated as potential limitations the Discussion, in particular on Aurora A localization and kinetochore-attachment lifetimes in higher eukaryotes. We would be happy to publish your paper in Life Science Alliance pending this change and final revisions necessary to meet our formatting guidelines.

MANUSCRIPT ORGANIZATION AND FORMATTING:

To avoid unnecessary delays in the acceptance and publication of your paper, please read the following information carefully. Full guidelines are available on our Instructions for Authors page, <https://www.life-science-alliance.org/authors>

- Please upload all figure files as individual ones, including the supplementary figure files; all figure legends should only appear in the main manuscript file
- Please add ORCID ID for corresponding author - you should have received instructions on how to do so.
- Please add the X and Bluesky handles of your host institute/organization, as well as your own, and/or one of the authors, in our system.
- It is recommended to exclude figures from the manuscript text and upload them separately.
- LSA does not permit citation of "data not shown," "manuscript in preparation," "manuscript submitted," etc., in any section of the manuscript.
- Please add your main and supplementary figure legends to the main manuscript text after the references section.
- Please rename "Competing Interest Statement" to "Conflict of Interest."
- Supporting references should be part of the main manuscript references list.
- Please upload your Tables in editable .doc or Excel format.
- The contributions selected for Peng-Ye Wang do not qualify them for authorship. Please either update the contributions in our system and in the Author Contributions section of the manuscript, or let us know if the author needs to be removed (and added potentially to the acknowledgment section).
- Please add an Author Contributions section to your main manuscript text.
- Please add callout for Figures 2A-B; 3A-B and S3A-B to your main manuscript text.
- Please remove the section "Online supplemental material" and include supplemental figure legends with the main figure legends.

We welcome submissions of potential cover images for the issue of LSA in which your work would appear. If you have high quality images associated with this work, please feel free to email these, with a caption, to the journal office.

LSA encourages authors to provide a 30-60 second video where the study is briefly explained. We will use these videos on social media to promote the published paper and the presenting author (for examples, see <https://docs.google.com/document/d/1-UWCfbE4pGcDdcgzcmiuJI2XMBJnxKYeqRvLLrLSo8s/edit?usp=sharing>). Corresponding or first-authors are welcome to submit the video. Please submit only one video per manuscript. The video can be emailed to contact@life-science-alliance.org

FINAL FILES:

The following items are required for acceptance.

The license to publish form must be signed before your manuscript can be sent to production. A link to the license to publish form will be available to the corresponding author only. Please take a moment to check your funder requirements.

Thank you for your attention to these final processing requirements. Please revise and format the manuscript and upload materials as soon as you are able.

Thank you for this interesting contribution to the literature. We look forward to publishing your paper in Life Science Alliance.

Sincerely,

Reviewer #1 (Comments to the Authors (Required)):

The authors have adequately addressed my concerns.

Reviewer #2 (Comments to the Authors (Required)):

The revised manuscript by Wang et al. only addresses concerns raised by myself and Reviewer #1 to a limited extent. The major concerns are: 1) the model is overly simplistic and does not take into account complexity observed within the cell (e.g., Aurora B localization, microtubule organization within the spindle, chromosome oscillations); and 2) many of the key conclusions have been reached through experimental work. The authors address these concerns primarily through text alterations and do not do a good job of conveying that the process of error correction is more complicated than the stated model. Unfortunately, my disposition continues to be that this paper is not appropriate for publication in Life Science Alliance in its current form.

April 16, 2026

RE: Life Science Alliance Manuscript #LSA-2025-03612-TRR

Dr. Ping Xie
Institute of Physics
Laboratory of Soft Matter Physics
Chinese Academy of Sciences, Beijing 100080
Beijing 100190
China

Dear Dr. Xie,

Thank you for submitting your Research Article entitled "An optimum rate of microtubule flux for error correction in metaphase spindle". It is a pleasure to let you know that your manuscript is now accepted for publication in Life Science Alliance. Congratulations on this interesting work.

Your article will publish open access upon publication under a CC-BY license.

DISTRIBUTION OF MATERIALS:

Again, congratulations on a very nice paper. I hope you found the review process to be constructive and are pleased with how the manuscript was handled editorially. We look forward to future exciting submissions from your lab.

Sincerely,
